



# On-line determination of the chemical composition of single activated cloud condensation nuclei – a first investigation of single urban CCN and CCN obtained from sea water samples

Carmen Dameto de España[1], Anna Wonaschuetz[1], Gerhard Steiner[1,3], Harald Schuh[1] , Constantinos Sioutas[2], Regina Hitzenberger[1]

[1]Aerosol Physics and Environmental Physics, Faculty of Physics, University of Vienna, 1090 Vienna, Austria
[2]Department of Civil and Environmental Engineering, University of Southern California, Los Angeles, 3620, USA
[3]Grimm Aerosol Technik Ainring, 83404 Ainring, Germany

*Correspondence to*: Carmen Dameto de España (carmen.dameto@univie.ac.at)

**Abstract.**

Cloud condensation nuclei (CCN) play an important role in cloud microphysics and are crucial for the second indirect effect of aerosols on global climate. One of the uncertainties in calculations of the indirect effect is due to insufficient data on CCN activation. The formation and growth processes of aerosol particles which subsequently become CCN determine their chemical composition. Due to the numerous organic and inorganic components present in atmospheric aerosol particles, a determination of the chemical composition of individual CCN is still challenging. To expand our understanding of activation of real-world CCN we introduce a novel method to characterize the chemical composition of single activated CCN in their droplet state. This method consists of a coupling of two essential instruments, a CCN-VACES (Cloud Condensation Nuclei-Versatile Aerosol Concentration Enrichment System) which is a modification of the original VACES to select and enrich CCN concentrations, and a Laser Ablation Aerosol Particle Time of Flight mass spectrometer (LAAPTOF), a single particle mass spectrometer. In the CCN-VACES, an aerosol flow is exposed to a specific water vapour supersaturation (in this study: 0.035 %, 0.054 %, 0.1 % and 0.6 %, respectively) and the CCN in the flow grow to droplets if their critical supersaturation is exceeded. These grown droplets are subsequently enriched in concentration by means of a virtual impactor at the end of the growth region by a factor of ca. 16 and pass directly into a LAAPTOF to measure the chemical composition of individual activated droplets. Contrary to widely held beliefs, the LAAPTOF is able to analyse refractory and non-refractory components even in aqueous droplets and can therefore be used to determine the chemical composition of actually activated CCN in their droplet state. Single particle spectra (for both positive and negative ions) were obtained from activated CCN in the ambient aerosol as well as activated CCN originating from aerosolized sea water samples collected at two different regions (Palma de Mallorca and San Sebastián, Spain). Ambient CCN were found to contain sometimes highly complex mixtures of different carbonaceous and non-carbonaceous components. Sea water derived CCN show the expected content of sea salt constituents, but the presence of organics is also observed. Activated CCN from the San Sebastián water samples have stronger sulphate





signals than the Mallorca water sample. The LAAPTOF was found to provide insights into the composition of individual activated CCN.

## 1 Introduction

Atmospheric aerosols have a strong influence on climate directly by interacting with incoming solar radiation and indirectly by acting as Cloud Condensation Nuclei (CCN) and ice nuclei (IN). CCN and IN can alter cloud properties such as albedo and cloud life time (e.g. Albrecht, 1989, Rosenfeld et al., 2014 and Twomey, 1977). The ability of aerosol particles to act as CCN depends on their size and chemical composition and the water vapour supersaturation of their environment (e. g. Dusek et al., 2006, McFiggans et al., 2006, Burkart et al., 2011, Burkart et al., 2012). Numerous studies indicate that chemical composition

has a strong influence on CCN activity (e. g. Hudson, 2007; Furutani et al., 2008; Quinn et al., 2008). In particular McFiggans et al. (2006) stressed the high relevance of the composition of ambient aerosol particles in the size range from 40 to 200 nm for CCN activation at the typical low supersaturations present at cloud formation. Particles smaller than 40 nm are too small and therefore unlikely to become activated regardless of their chemical composition, while particles with sizes larger than 200 nm will usually contain enough soluble material to activate into cloud droplets. Several laboratory studies characterised the

activation of well-known single chemical component aerosol particles (e. g. Bilde and Svenningsson, 2004, Giebl et al., 2002, Burkart et al., 2011). Single ambient aerosol particles contain on the order of $10^2$ to $10^{15}$ molecules per particle and have masses $\sim 10^{-20}$ to $10^{-6}$ g (Pratt and Prather, 2012). They often consist of internal mixtures of possible unknown organic and inorganic components (e. g. Murphy et al., 2006; Friedman et al., 2013, Okada et al. 2001). In contrast to the inorganic aerosol components which consist of mostly a few well characterised compounds, the organic fraction of the aerosol material

comprises hundreds of individual species (Kanakidou et al., 2005). Water soluble organic carbon (WSOC) has been shown to influence particle activation (e.g. McFiggans et al., 2006, Jacobson et al., 2000).

Ambient aerosols originate from multiple different sources. Chemical reactions of natural and/or anthropogenic precursor gases lead to particle nucleation events. The freshly formed secondary aerosol particles can attain CCN ability during

atmospheric ageing processes (e.g. Asmi et al., 2011, Dameto de España et al., 2017, Németh et al., 2018). Natural sources contribute ca. 90 % by mass to atmospheric aerosols, with sea salt aerosol (SSA) and dust as the largest fractions (Prather et al., 2013). SSA properties are still not well characterized. Several studies reveal that SSA are complex mixtures of inorganic sea salt and organic compounds with different solubilities (e.g. Prather et al., 2013; Blanchard et al., 1989; Parungo et al., 1986, Middlebrook et al., 1998). The role of the contribution of organic material to SSA in remote regions is still uncertain

(O´Dowd and de Leeuw 2007). Models predict that organic matter can enhance cloud droplet concentrations (O´Dowd et al., 2004) under the assumption that SSA particles are internally mixed. Recent studies (e.g. Leck et al., 2005; Pratt et al., 2009) indicated however, that some SSA particles smaller than 200 nm could be externally mixed. As SSA is such a major component



of the atmospheric aerosol and provides CCN over large areas of the globe, a rigorous investigation of SSA characteristics is necessary.

In addition, the physical and chemical processes that occur during atmospheric ageing of particles continuously change their properties. Offline bulk measurements to determine the chemical composition do not have sufficient temporal resolution to characterize the dynamic changes in the composition of ambient aerosol particles. Measurements of the chemical composition of single ambient particles are still challenging. To address this challenge on-line mass spectrometry techniques have been intensively developed over the last decade.  As opposed to offline techniques they can provide information of chemical changes

in atmospheric aerosol particles on short time scales (Pratt and Prather, 2012). The most widely used on-line techniques are single particle laser ablation (Noble and Prather, 2000) and the Aerodyne aerosol mass spectrometer (AMS) which both have been widely used to analyse the bulk chemical composition of ambient aerosols (Canagaratna et al., 2007).

Laser desorption/ionisation (LDI) is currently used for single particle analysis. In contrast to the AMS, the Single Particle Mass

Spectrometer (SPMS) (Johnston et al., 2000, Hinz and Spengler, 2007) is able to analyse both non refractory (e.g. organics, ammonium nitrate) and refractory (e.g. mineral dust, soot) components of single atmospheric aerosol particles (Pratt and Prather, 2012). Cziczo et al. (2003) and Cziczo et al.  (2006) coupled a continuous flow ice nuclei counter (Rogers et al., 2001) to a single particle mass spectrometer (PALMS; Murphy et al, 1998) to focus on chemical characterization of IN, and single particle analyses of ice particle residuals where conducted by Schmidt et al (2017) at Jungfraujoch.


The LAAPTOF is a recent commercially available single particle mass spectrometer manufactured by AeroMegt GmbH, which has been used already in several studies, such as Shen et al. (2018b) who studied atmospheric particles, Mardsen et al. (2018) who studied mineral phases in dust aerosol particles, while Wonaschütz et al. (2017) characterized particles formed radiolytically in aerosol neutralizers. Ahern et al. (2016) compared the response of the LAAPTOF and another commercial

single particle mass spectrometer, an infrared (IR) laser vaporization soot-particle aerosol mass spectrometer (SP-AMS, Aerodyne Research Inc.) to secondary organic material condensing on biomass-burning soot particles and found that the LAAPTOF gives quantitative results for organic material even for complex aged biomass burning particles. Weiss et al. (2018) examined particles emitted by four mastic asphalt mixtures at different temperatures (195 °C to 245 °C) using the LAAPTOF.

All these studies performed with the various types of single particle mass spectrometer analysed dry particles. In the atmosphere, however, aqueous aerosols play an important role especially in fog, haze or cloud processes. The presence of water in aerosol samples, however, complicates particle detection by SPMS since the water readily evaporates in the vacuum inside the instruments leading to vacuum breakdown within the mass spectrometer hindering particle detection. If the droplets spend only a short time in the vacuum prior to analysis, SPMS measurements should be possible. Neubauer et al. (1997)

demonstrated that on-line laser desorption / ionization mass spectrometry is capable of analysing aqueous solutions.





A few studies directly measured the chemical composition of single particles in the size range critical to cloud formation. Zauscher et al. (2011) measured single particles in the size range 50 – 200 nm aerosol after growing them in a growth tube at high water vapour supersaturations (20%). Roth et al. (2010) analysed cloud residues and out-of-cloud aerosol particles with

diameters between 150 and 900 nm with an ALABAMA SPMS described by Brands et al. (2011). Hiranuma et al. (2011) separated droplets exiting from a cloud condensation nuclei counter from inactivated particles with a counterflow virtual impactor, dried them, measured the chemical composition of the resulting residual particles with a PALMS SPMS (Murphy et al, 1998), compared the chemical composition of inactivated particles and residuals and interestingly found higher sulphate signals in the dried residuals.

To the best of our knowledge, no study to-date measured the chemical composition of single activated CCN on-line in their unaltered droplet state; to address this need, we developed a method to characterize single droplets originating from CCN activated at different specific supersaturations. We utilized a new CCN-VACES (Versatile Aerosol Enrichment System; described by Dameto de España et al., 2019) in series with a LAAPTOF. The CCN-VACES is a modification of the original VACES (Kim et al., 2001a, Kim et al., 2001b) and is designed to activate CCN at low supersaturations (in our case 0.035,

0.054, 0.1 and 0.06 %) and enrich the concentration of the grown droplets for further analysis. The goals of this study are to analyse the chemical composition of single ambient CCN and of CCN formed from sea water samples by activating them at different supersaturations and analyse directly the grown droplets in their liquid state.

## 2 Instrumentation

### 2.1 CCN-VACES

The CCN-VACES is a modified version of the VACES (Kim et al., 2001a, Kim et al., 2001b) designed to enrich CCN concentrations. First descriptions of this modified system were shown by Dameto de España (2018a, 2018c and 2019b). A full description and characterisation of the instrument was published recently (Dameto de España et al., 2019). Briefly, the CCN-VACES consists of two main parts, a saturator and two parallel condenser tubes each followed by a virtual impactor with cut size of 1.5 µm. (For the experiments described in this study, only one condenser tube with its virtual impactor was used. The

other tube also received the aerosol flow but the flows of the virtual impactor connected to this tube were not used for the measurements.)

The saturator is a cylindrical tank half filled with ultrapure water (Direct-Q5®, Millipore, Billerica, MA) heated with two hotplates (1500 W, EKP3582, Claronc®). Water temperature is kept constant (usually at 52°C) within ±0.1°C. The temperature profile in the water tank is homogenized with a peristaltic pump and a temperature sensor measures the water temperature $T_w$

within ±0.1 °C. The top cover of the water tank is connected to the two condenser tubes each composed of two concentric tubes. An ethylene glycol/water (1:1 by volume) coolant circulates between these concentric tubes to establish the water vapour supersaturation inside. The coolant temperature, $T_c$ is regulated with a chiller (Thermocube 300 1D 1 LT Solid State Cooling Systems, Pleasant Valley, NY).



In the virtual impactors after the condenser tubes, grown droplet particles with sizes above the cut size are concentrated into the minor flow (5 l/min) with an enrichment factor of ca. 16 (Dameto de España et al., 2019). The major flow (100 l/min) containing only particles with smaller sizes is removed from the system. As described in previous studies (e.g. by Geller et al., 2004 or Sioutas et al., 1998) the performance of the virtual impactor is defined by the major and the minor flows. Therefore, the flow rates were always checked before each of the measurements. The minor flow of the virtual impactor containing the
activated droplets as well as particles with sizes below the cut size represents 5% of the total flow. As the LAAPTOF has a lower size detection limit of 300 nm and as atmospheric aerosol particles with sizes between 300 nm and 1.5 µm that remain inactivated at the supersaturations used here are very rare, this small percentage will likely not influence the LAAPTOF measurements unduly.

In order to enrich CCN concentrations the aerosol passes first through the saturator region with 100% relative humidity and then to the condenser tube. The duct connecting the saturator and the condenser was heated with a heating wire to avoid premature water condensation. Before entering the condenser tube the air temperature (saturator temperature $T_s$) is measured. In the condenser tube the aerosol flow is cooled down, the water vapour becomes supersaturated and particle activation and growth take place. The temperature of the aerosol flow exiting the condenser is measured ($T_{out}$). The whole system is controlled
by four different temperature sensors and the supersaturation is set by adjusting the temperatures according to a calibration curve as described by Dameto de España et al. (2019).

### 2.2 LAAPTOF (Laser Ablation Aerosol Particle Time Of Flight) mass spectrometer

The on-line aerosol single particle mass spectrometer LAAPTOF (AeroMegt GmbH) provided the chemical information for the activated droplets. The LAAPTOF is a recent commercially available SPMS able to analyse refractory and non-refractory
components (Gemayel et al., 2016). Exhaustive characterizations of the instrument were performed by Gemayel et al. (2016), Marsden et al. (2016) and Ramisetty et al. (2018). Several studies have already described and characterized LAAPTOFs (e.g. Wonaschuetz et al., 2017, Gemayel et al., 2016; Shen et al., 2018a, Ramisetty et al., 2018, Mardsen et al., 2016). In brief, the LAAFTOF comprises three different regions as shown in Fig. 1. The first part is the inlet which consists of a 100 µm critical orifice and an aerodynamic lens designed by Liu et al. (1995a, 1995b) that focuses particles into a narrow beam by passing the
flow through six different apertures with decreasing diameters and transmits the particles into the vacuum stage. The sampling flow is 0.1 l/min. Particles with sizes between 70 nm and 2.5 µm vacuum aerodynamic diameter ($d_{va}$) are transmitted into the vacuum stage (pressure 10$^{-7}$ mbar) with 100% efficiency (AeroMegt Usermanual, 2015) (in our setup, however, spectra are obtained only for particles >300 nm). This second region, the Particle Time of Flight (PTOF) region, is responsible for particle sizing. It consists of two 50 mW laser diodes (wavelength 405 nm) operating in a continuous wave mode, separated by 11.3
cm. The two lasers irradiate the particles and as a particle successively crosses the two laser beams, the two scattering signals are recorded and its flight time is determined. The vacuum aerodynamic diameter of a particle is derived from its flight time





from a calibration with particles of well-known sizes. In our case we used polystyrene latex particles PSL particles (PSL, Polyscience Inc. Warrington, PA) with sizes between 350 nm and 1500 nm. After the second sizing laser, laser ablation takes place in the $5*10^{-7}$ mbar vacuum. An excimer laser (wavelength 193nm; ATLEX 300 I, ATL Lasertechnik GmbH) with a

maximum energy of 10 mJ per pulse (10 ns) allows ablation of a single particle every 4 µs (Shen et al., 2018a). The resulting ions and particle fragments are extracted in the third region into the positive and negative time of flight mass spectrometer (Tofwerk, TOF). Charged ions are detected independently by two microchannel plate detectors (MCP). A first pilot study on the ability of the LAAPTOF to detect and analyse aqueous droplets was performed by Dameto de España et al. (2018b).

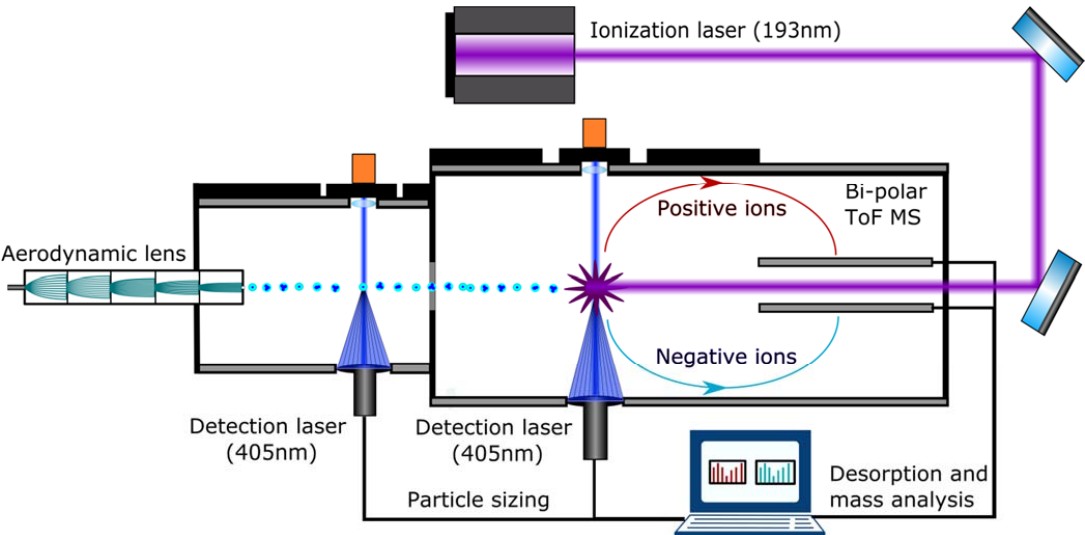

**Figure 1: Schematic diagram of the LAAPTOF.**

**3 Experimental set-up**

The chemical composition of activated CCN was obtained by conducting measurements with the CCN-VACES coupled to the LAAPTOF. As the LAAPTOF inlet flow is only 0.1 l/min, the minor flow from the virtual impactor of the CCN-VACES (5 l/min) was split into three flows. One flow (4.3 l/min, regulated with a flow-controller MC-20SLPM-D, Alicat Scientific, Inc.)

was drawn with a pump and vented to the outside, 0.6 l/min flowed through a diffusion dryer and then to a condensation particle counter CPC (Grimm 5412, flow regulated by a critical orifice) to check the aerosol number concentration and the remaining 0.1 l/min flow entered the LAAPTOF. An illustration of the experimental set-up is shown in Fig. 2.


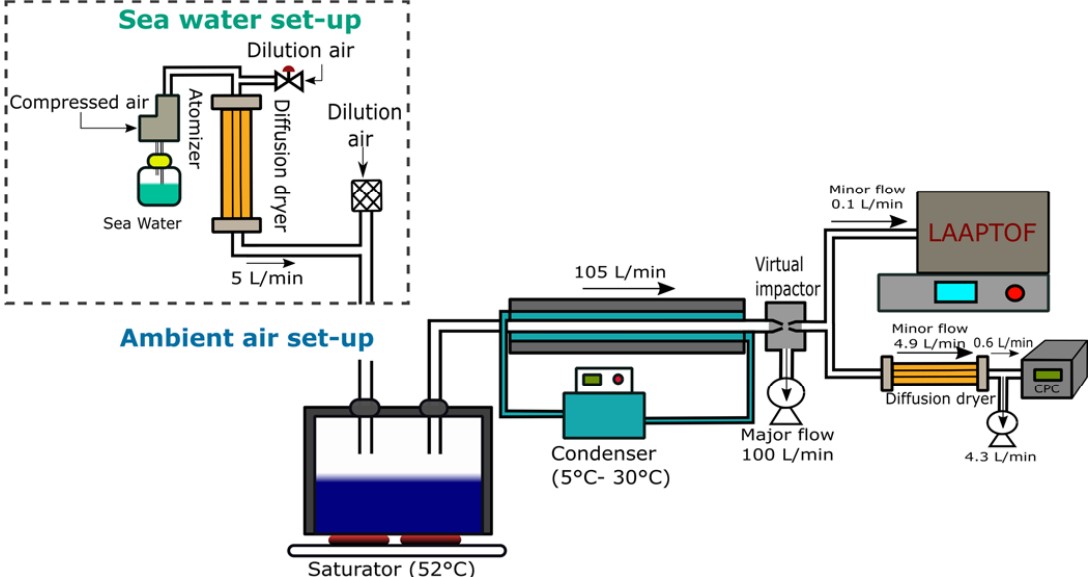

**Figure 2: Schematic illustration of the experimental set-up. For ambient air sampling the inlet was open. For the sea water**
**experiment the set-up in the dashed square was coupled to the inlet.**

## 4 Measurements

### 4.1 LAAPTOF calibration and measurements

The Time-of-Flight mass spectrometer provides a set of peaks which have to be attributed to ion masses. A basic calibration
was performed using Carbon Black (Elftex 124, Cabot Corporation) particles produced by aerosolizing a suspension in an
80/20 water/isopropanol mixture. The resulting single particle spectra contain only peaks for carbon ions, which can easily be
attributed to single C atoms and multiples ($C_n$ ions Fig. 3). This basic calibration is then used for the identification of ion
masses from unknown ionic fragments arising from the laser ablation. Data evaluation and data processing was performed
with LAAPTOF Data Analysis© software (Copyright AeroMegt GmbH 2014).





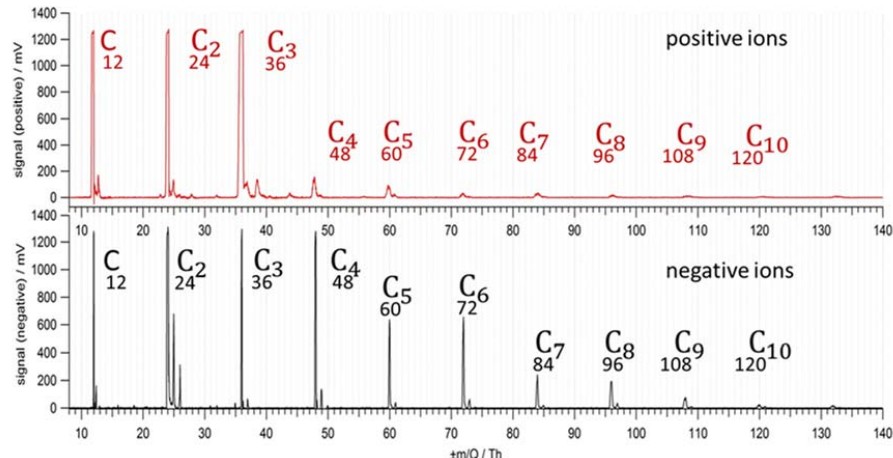


**Figure 3: Single particle spectra of back carbon aerosol**

### 4.2 Measurements of ambient air CCN

These CCN measurements were performed using the set-up shown in Fig. 2. Ambient air was drawn directly into the saturator

and the water temperature was set to 52°C. The temperature in the condenser tubes was adjusted according to the desired $T_{out}$ which is related to the water vapour supersaturation in the flow exiting the tube before entering the virtual impactor. Details of setting a desired supersaturation can be found in the previous paper by Dameto de España et al. (2019). In the supplementary data, Table S1 provides the temperature settings in the CCN-VACES and the corresponding supersaturations used in our study.

### 4.3 Measurements of CCN produced from sea water samples

Seawater samples were collected in two different locations in Spain; one on 06.06.2018 in Palma de Mallorca (39°33'21.8"N, 2°40'50.0"E) and the second on 21.06.2018 in San Sebastián (43°18'53.9"N 1°59'43.5"W). The samples were kept frozen until the measurements were conducted. Particles were generated by aerosolizing the seawater with a Collison atomizer (TSI, 3076) operated with particle free air at 1.2 bar producing a 2 L/min aerosol flow. This flow was diluted with 3 L/min of dry clean air, dried to a relative humidity < 15% with a diffusion dryer and then introduced into the CCN-VACES. The condenser

temperature was chosen according to the desired water supersaturation at the end of the condenser tube.



## 5 Results

### 5.1 Results of analyses of activated ambient air CCN droplets

Ambient air CCN were activated in the CCN-VACES and introduced into the LAAPTOF to obtain single particle spectra for each of the droplets. Measurements were performed on five different days with laboratory ambient air at four different
supersaturations. The number of spectra obtained on each day and supersaturation are represented in Table 1. The total number of spectra obtained e.g. at a supersaturation of 0.6% varies between 3150 (25.03.2019) and 10000 (30.04.2019). The number of spectra obtainable within one measurement series was limited either by the pressure increase in the vacuum chamber caused by the evaporating droplets or the limits imposed by the instrument software.

In analysing dry aerosol particles, ca. 80% of the obtained spectra give sufficiently clear peaks for data evaluation. As shown by Neubauer et al. (1997), particle water content influences particle detection in SPMS. In our case of analysis of activated CCN (droplets), only ca. 10% to 20% of the spectra obtained from the droplets were sufficiently clear for analysis, which means >300 spectra for each set of measurements (date, supersaturation) are available for analysis, a number sufficient for statistical data processing. Hinz et al. (2004) set the lower limit for the number of spectra at 200.


A specific mass calibration was performed selecting three different marker peaks (last two columns in Table 1) from the positive and negative spectra. Relevant marker peaks observed in the positive ions spectra are mainly those of carbon ions ($C^+$, $C^+_2$, $C^+_3$, m/z= 12, 24 and 36), potassium ($K^+$, m/z= 39 and 41), $Si^+$ (m/z= 28) and $NO^+$ (m/z= 30). In the negative spectra, $O^-$ and $OH^-$ (m/z= 16 and 17), $C^-_2$, $C^-_3$, $C^-_4$ (m/z= 24, 36 and 48) and $HSO^-_4$ (m/z= 97) predominate. As suggested by Shen et al.
(2018a), peaks were chosen with a wide separation between peaks in the negative spectra. In the positive spectra peaks were selected for the specific mass calibration that gave the strongest signals.







| Day | SS (%) | Total number of spectra | Number of spectra used | Fraction of Spectra used for analysis (%) | Positive peaks for calibration (m/z) | Negative peaks for calibration (m/z) |
|---|---|---|---|---|---|---|
| 25.03.2019 | 0.035 | 10000 | 1199 | 12% | 12/24/36 | 24/36/48 |
| 25.03.2019 | 0.054 | 10000 | 1007 | 10% | 12/24/36 | 16/24/97 |
| 25.03.2019 | 0.6 | 3150 | 688 | 22% | 12/36/39 | 24/36/72 |
| | | | | | | |
| 29.04.2019 | 0.035 | 3150 | 342 | 11% | 12/39/41 | 16/80/97 |
| 29.04.2019 | 0.054 | 3090 | 274 | 9% | 12/36/39 | 16/80/97 |
| 29.04.2019 | 0.1 | 3100 | 368 | 12% | 12/28/30 | 16/80/97 |
| 29.04.2019 | 0.6 | 7160 | 834 | 12% | 18/24/30 | 17/80/97 |
| | | | | | | |
| 30.04.2019 | 0.035 | 5110 | 606 | 12% | 12/30/36 | 17/80/97 |
| 30.04.2019 | 0.054 | 5330 | 839 | 16% | 12/30/36 | 17/80/97 |
| 30.04.2019 | 0.1 | 5130 | 803 | 16% | 12/28/30 | 16/80/97 |
| 30.04.2019 | 0.6 | 10000 | 1556 | 16% | 12/36/39 | 17/24/26 |
| | | | | | | |
| 15.05.2019 | 0.035 | 4090 | 449 | 11% | 12/30/36 | 17/80/97 |
| 15.05.2019 | 0.054 | 3990 | 555 | 14% | 12/30/36 | 17/80/97 |
| 15.05.2019 | 0.1 | 4170 | 630 | 15% | 12/28/30 | 16/80/97 |
| 15.05.2019 | 0.6 | 4170 | 558 | 13% | 12/36/39 | 17/24/26 |
| | | | | | | |
| 16.05.2019 | 0.035 | 4060 | 526 | 13% | 12/30/36 | 17/80/97 |
| 16.05.2019 | 0.054 | 3480 | 410 | 12% | 12/30/36 | 17/80/97 |
| 16.05.2019 | 0.1 | 4130 | 558 | 14% | 12/28/30 | 16/80/97 |
| 16.05.2019 | 0.6 | 4340 | 576 | 13% | 12/36/39 | 17/24/26 |

**Table 2: Summary of spectra obtained for activated ambient air CCN and strongest ion signal used for the specific mass calibration.**


For the evaluation of the large number of available spectra, automated data processing was performed with a fuzzy c-means clustering algorithm (Hinz et al., 1999) incorporated in the LAAPTOF Data Analysis Igor software (Version 1.0.2, AeroMegt GmbH). This algorithm groups single particle spectra in different clusters according to the similarity of the spectra (Reitz et al., 2016). Every single spectrum corresponds to a specific particle. This program, however, needs the number of clusters to

be selected as an input variable. In our study, we used another fuzzy c-means clustering program to obtain the number of clusters best representing the data (i.e. a measurement series) which was used then as input to the instrument software. The





membership parameter µ obtained from the fuzzy c-means clustering determines the degree a single particle spectrum belongs to each of the clusters according to their spectra similarity. These µ values vary from to 0 to 1, with 0 no pertinence and 1 full correspondence with the assigned cluster. Fuzzy c-means clustering enables to assign a particle to multiple clusters and the

assigned fraction to each cluster arises from the µ value. Every class is represented by a central spectrum, which provides a visualization of the predominant chemical signals in this cluster. Following Hinz et al. (2005), the number of particles corresponding to each cluster was determined by counting the number of particles with a membership µ value > 0.7 (Hinz et al., 2005).

The data set for each day and each supersaturation was evaluated with the fuzzy c-means clustering using the methodology above. The activated CCN droplets measured on 25.03.2019, 29.04.2019 and 30.04.2019 can be grouped in four different clusters, which are referred to as "classes" in the following text. Activated CCN droplets measured on 15.05.2019 and 16.05.2019 were classified into only three classes, as on those days class 4 or "nitrate rich" (see below) was not observed. The number of particles corresponding to each class, which means particles with a µ factor > 0.7, are summarized in Table 2. For

the measurements at 0.1% water vapour supersaturation on 25.03.2019 no data are available due to a technical problem.

### 5.1.2 Classification of measured ambient air CCN spectra

The single droplet spectra were attributed to four classes to obtain a general impression of the chemical composition and predominant species and to enable comparisons with the results from other studies. Example of average spectra corresponding to the classes are illustrated in Figs. (4-7) which correspond to the measurements performed on 30.04.2019 at SS=0.6%


Spectra in the first class, class 1, or "**internally mixed**" spectra contain many different ions in the positive ion spectrum. Droplets in this class contain a variety of ions of different compounds or elements, which indicate that the original CCN had been internally mixed particles (Kane and Johnston, 2000). Secondary aerosol material is represented by the nitrosonium ion $(NO^+, m/z=30)$ and $NH_4^+$ (m/z=18) (see Brand et al., 2011, Hinz et al., 2006). The presence of carbonaceous species is

represented by $C_n^+$ (m/z= 12, 24 and 36). Sodium $Na^+$ (m/z=23), aluminium $Al^+$ (m/z=27), calcium $Ca^+$ (m/z=40), potassium $K^+$ (m/z=39, 41) and $Fe^+$ (m/z=56) ions are present in the spectrum. These latter peaks were also observed for dry particles analysed at Jungfraujoch in the Swiss Alps (3580 m a.s.l.) by Hinz et al. (2005). Organic particle fragments such as $C_2H_3^+$ (m/z=27) in the positive ions spectrum and $C_2H_2^-$ (m/z=26) in the negative ions spectrum indicate the presence of organic material (Brands et al., 2011).

In the negative ions spectrum signs of $O^-$ (m/z=16) and $OH^-$ (m/z=17) correspond to water. Ions of organic species are represented by $C_X^- H_Y^-$ (m/z= 25, 26 and 27). Ammonium nitrate (represented by $NO_2^-$, m/z=46 and $NO_3^-$, m/z=62) and sulphate (peaks for $HSO_4^-$, m/z=97, and $SO_3^-$, m/z=80) again show the presence of secondary material.





The second class, class 2, we called "**carbon and nitrate**". In this class strong elemental carbon peaks are observed in the positive spectrum as $C_n^+$ (m/z= 12, 24 and 36). Lower intensity peaks for inorganic elements can be distinguished. The nitrosonium ion $NO^+$ (m/z=30) associated to ammonium nitrate is also present as well as the siliceous ion $Si^+$ (m/z= 28), which is often observed in measurements of the chemical composition of the ambient aerosol (e.g. Gemayel et al., 2016, Brands et al., 2011). In the negative ions spectrum of this class, no noticeable peaks were observed.

The third class, class 3, called "**carbonaceous rich**", is characterized by dominant positive carbonaceous ions ($C^+$, $C^+_2$, $C^+_3$, m/z= 12, 24 and 36). In the negative ions spectrum, weak peaks related to ammonium sulphate $HSO_4^-$ (m/z=97) are observed.

The fourth class, class 4, named "**nitrate rich**", has a predominant peak of $NO^+$ (m/z=30) associated with nitrate. Carbon peaks with lower intensities can be observed in the positive ions spectrum ($C^+$, $C^+_2$, $C^+_3$, m/z= 12, 24 and 36) and $Si^+$ (m/z= 28) and $NH_4^+$ (m/z=18) are also present. In the negative ions spectrum, weak peaks related to ammonium sulphate $HSO_4^-$ (m/z=97) and ammonium nitrate, indicated by $NO_2^-$ (m/z=46), are observed. This "nitrate rich" class 4 is similar to class 2 "carbon and nitrate", but the intensities of the peaks differ. In class two, peaks of carbonaceous ions are stronger whereas in class four the nitrate peaks are more prominent.

Table 2 shows spectra obtained from ambient air CCN activated at different supersaturations, the attribution of spectra to classes 1 – 4 in terms of number of individual spectra per class as well as the percentage of total spectra obtained during the measurement series.

| 25.03.2019 | class 1 (# spectra) | class 2 (# spectra) | class3 (# spectra) | class 4 (# spectra) | TOTAL (# spectra) | class 1 (%) | class 2 (%) | class3 (%) | class 4 (%) |
|---|---|---|---|---|---|---|---|---|---|
| SS=0.035% | 109 | 469 | 164 | 221 | 963 | 11% | 49% | 17% | 23% |
| SS=0.054% | 50 | 213 | 338 | 177 | 778 | 6% | 27% | 43% | 23% |
| SS=0.6% | 51 | 284 | 57 | 172 | 564 | 9% | 50% | 10% | 30% |
| **29.04.2019** | | | | | | | | | |
| SS=0.035% | 45 | 100 | 69 | 63 | 277 | 16% | 36% | 25% | 23% |
| SS=0.054% | 23 | 101 | 43 | 43 | 210 | 11% | 48% | 20% | 20% |
| SS=0.1% | 140 | 30 | 54 | 224 | 448 | 31% | 7% | 12% | 50% |
| SS=0.6% | 47 | 352 | 143 | 81 | 623 | 8% | 57% | 23% | 13% |
| **30.04.2019** | | | | | | | | | |
| SS=0.035% | 36 | 160 | 205 | 97 | 498 | 7% | 32% | 41% | 19% |
| SS=0.054% | 47 | 280 | 239 | 111 | 677 | 7% | 41% | 35% | 16% |
| SS=0.1% | 43 | 270 | 241 | 115 | 669 | 6% | 40% | 36% | 17% |
| SS=0.6% | 83 | 464 | 500 | 215 | 1262 | 7% | 37% | 40% | 17% |
| **15.05.2019** | **class 1** | **class 2** | **class3** | | **TOTAL** | **class 1** | **class 2** | **class3** | |
| SS=0.035% | 41 | 130 | 176 | | 347 | 12% | 37% | 51% | |
| SS=0.054% | 47 | 150 | 241 | | 438 | 11% | 34% | 55% | |
| SS=0.1% | 56 | 133 | 330 | | 519 | 11% | 26% | 64% | |
| SS=0.6% | 54 | 274 | 145 | | 473 | 11% | 58% | 31% | |
| **16.05.2019** | | | | | | | | | |
| SS=0.035% | 32 | 240 | 150 | | 422 | 8% | 57% | 36% | |
| SS=0.054% | 33 | 183 | 125 | | 341 | 10% | 54% | 37% | |
| SS=0.1% | 53 | 171 | 223 | | 447 | 12% | 38% | 50% | |
| SS=0.6% | 67 | 237 | 166 | | 470 | 14% | 50% | 35% | |

Table 2: Summary of the number of particles with a μ factor > 0.7 for a class





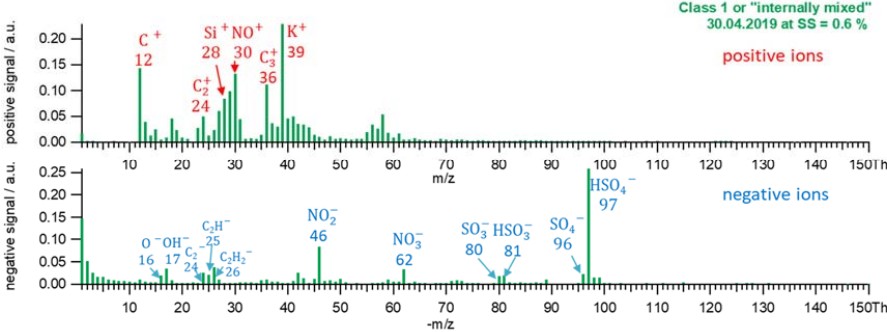

**Figure 4: Central spectrum corresponding to class 1 or "internally mixed"**

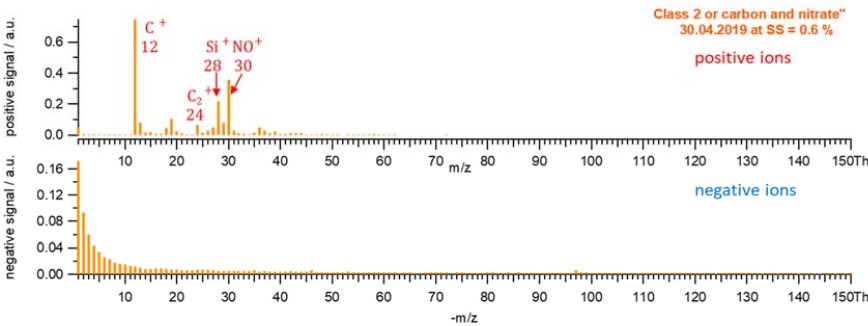

**Figure 5: Central spectrum corresponding to class 2 or "carbon and nitrate"**

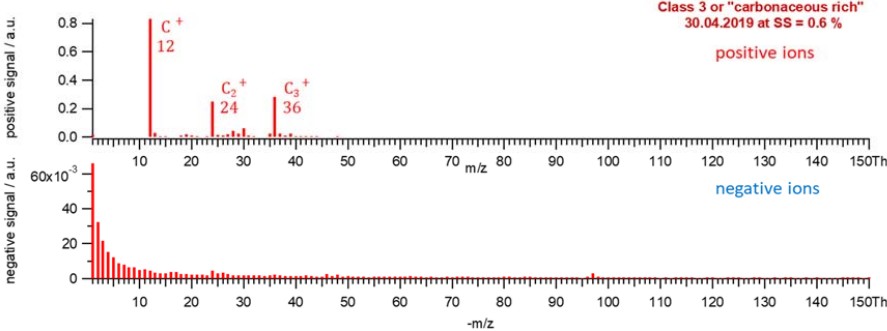

**Figure 6: Central spectrum corresponding to class 3 or "carbonaceous rich"**



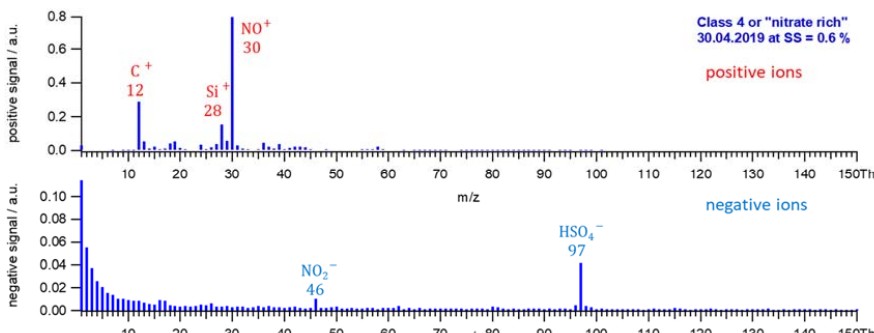

**Figure 7: Central spectrum corresponding to class 4 or "nitrate rich"**

In the analysis of ambient aerosol CCN, spectra belonging to classes 2, 3 and 4 appear for activated CCN droplets at all

supersaturations with small differences in peaks' intensity. In contrast, spectra attributed to class 1 ("internally mixed") show quite some variation. Fig. 8 shows examples of spectra obtained at each of the four supersaturations for particles belonging to to class 1 or "internally mixed". These spectra were obtained by setting the number of clusters in the fuzzy clustering routine to 1 for the measurements on 30.04.2019, so all spectra are grouped together. In the positive spectrum carbon peaks intensities ($C^+$, $C^+_2$, $C^+_3$, m/z= 12, 24 and 36) are higher for 0.054% supersaturation. The nitrosonium ion ($NO^+$, m/z=30) signal is stronger

at 0.1% supersaturation, while the potassium $K^+$ (m/z=39, 41) signal is stronger at 0.054% supersaturation. The peak intensities for organic negative ions $C^-_X H^-_Y$ (m/z= 25, 26 and 27) are stronger at 0.054% supersaturation. Ammonium nitrate ($NO^-_2$, m/z=46, and $NO^-_3$, m/z=62) is present at all supersaturations. The most prominent peak is from ammonium sulphate ($HSO^-_4$, m/z=97) for all supersaturations.

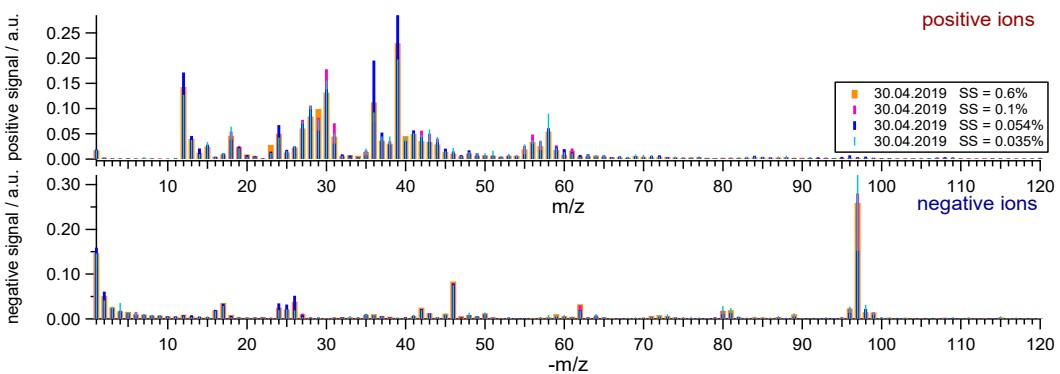

**Figure 8: Superposition of spectra belonging to class 1 at different supersaturations measured on 30.04.2019**



### 5.1.3. Spectra obtained from single droplets

The fuzzy c-means clustering is a commonly used method to analyse large numbers of spectra and to give an overview of the composition of different kinds of particles. This method, however, provides only an average spectrum of the single spectra, so

information on individual particles is lost. The major advantage of the link of CCN-VACES and LAAPTOF, however, lies in the ability to obtain spectra of single activated CCN. Therefore single spectra corresponding to each of the classes obtained from ambient CCN activated at 0.6% supersaturation on 25.03.2019 were analysed.

Figure 9 shows a single drople spectrum contained in the class "carbon and nitrate" indicating an internally mixed CCN containing an appreciable amount of Carbon.


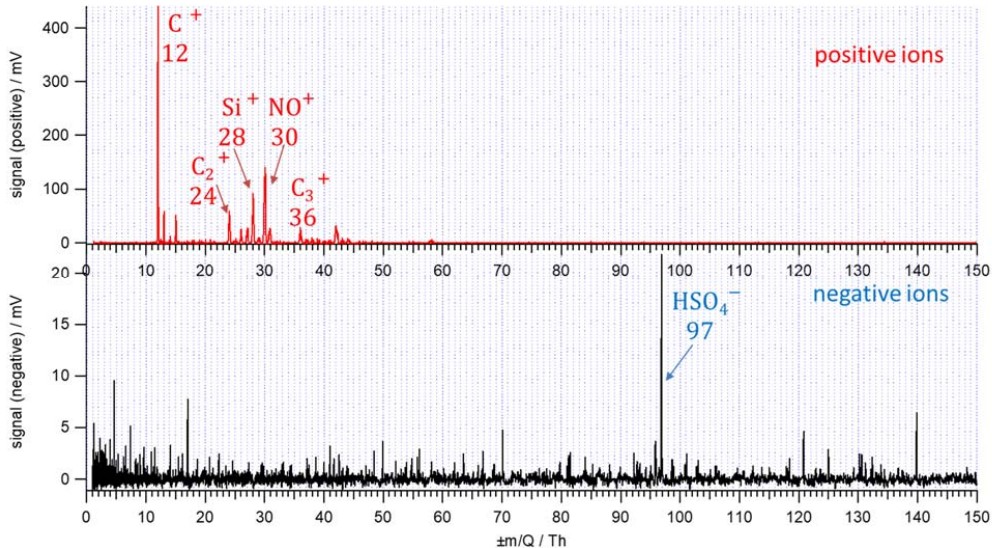

**Figure 9: Single particle spectra corresponding to class 2 or "carbon and nitrate"**

Figure 10 shows a spectrum of a droplet corresponding to the class 3 or "carbonaceous rich". Of particular note is that the average negative ions spectrum for this class does not show distinct peaks, while in the single droplet spectrum clear single carbon C and multiples' ($C_n$) peaks can be observed (C to $C_8$, m/z= 12 to 96) in both the positive and negative ions spectrum. In the positive spectrum more multiple C peaks are observed in the single droplet spectrum than in the mean spectrum corresponding to this class. This spectrum indicates that the initial CCN was a black carbon particle – possibly a freshly emitted

Diesel soot particle.





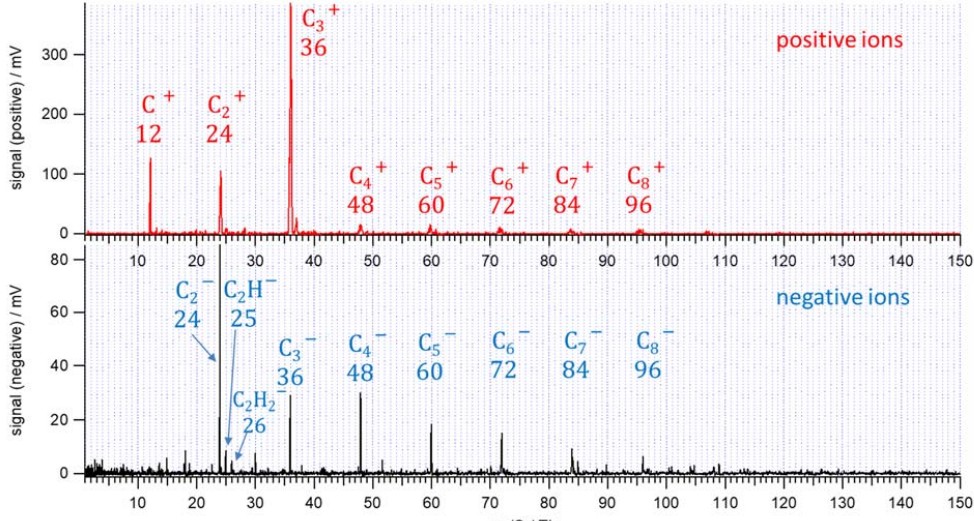

**Figure 10: Single particle spectra of single particle back carbon aerosol corresponding to the class 3 or "carbonaceous rich"**

Figure 11 gives the spectrum of a single droplet corresponding to the class "nitrate rich". Strong peaks in the negative spectra can be observed.

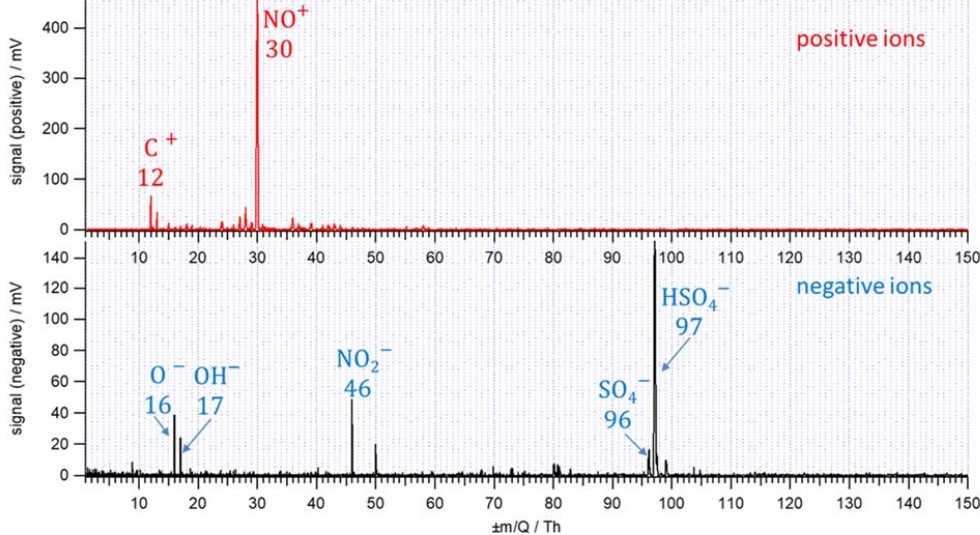


**Figure 11: Single particle spectrum corresponding to class 2 or "carbon and nitrate"**





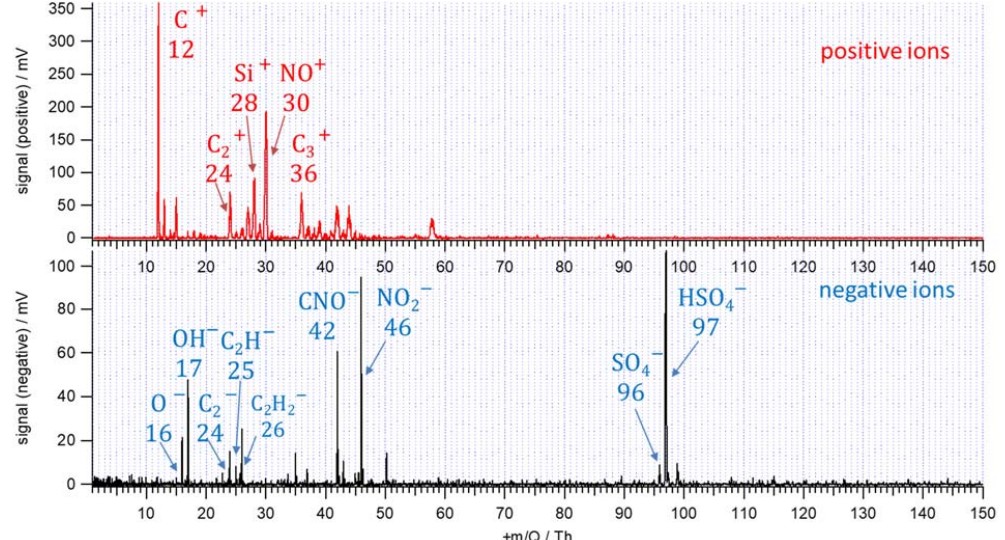

**Figure 12: Single particle spectrum corresponding to class 1 or "internally mixed"**


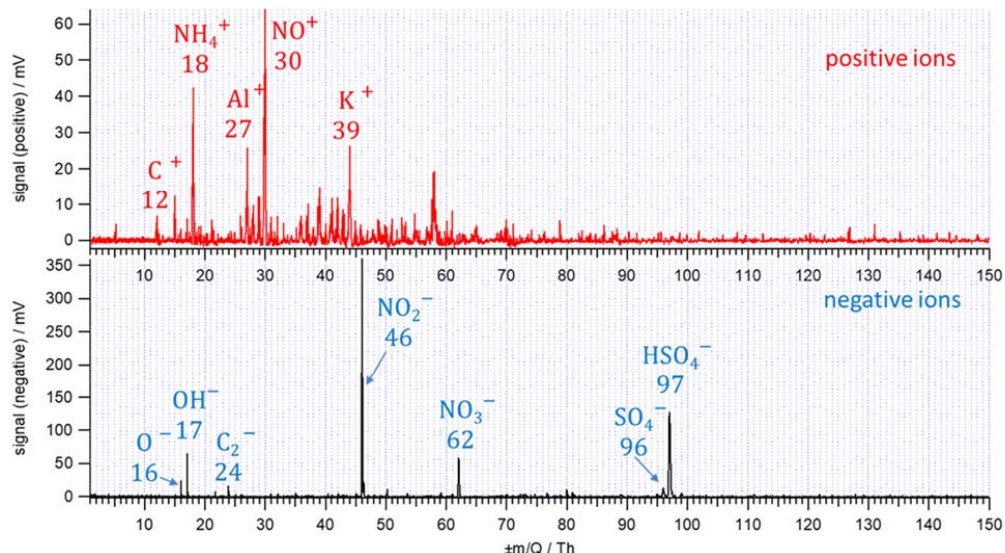

**Figure 13: Single particle spectrum corresponding to class 1 or "internally mixed"**



Figures 12-15 show droplet spectra corresponding to the class internally mixed. The presence of Si$^+$ (m/z= 28) in the spectrum in Fig.12 might indicate a CCN not originating in the area of Vienna. Okada and Hitzenberger (2001), e. g. found that the

presence of fine (< 2 μm) Si in the Vienna aerosol occurred for air mass origins over Upper Silesia (Poland). The original CCN of the droplet shown in Figure 13 may have had its origin in biomass combustion as indicated by the strong K peak (Pachon et al., 2013). Aluminium and NH$_4^+$ are also detected in this CCN.

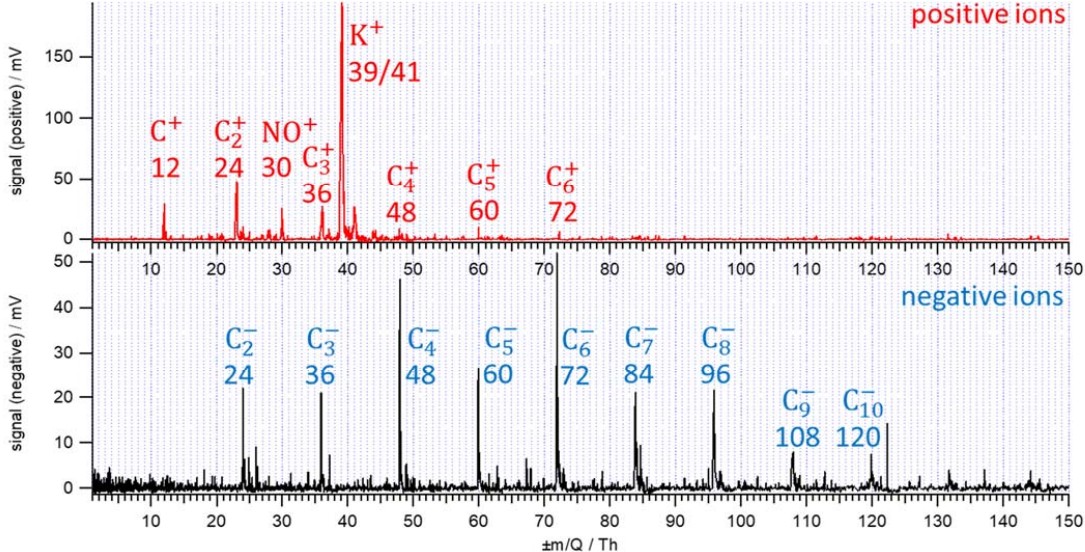

**Figure 14: Single particle spectrum corresponding to class 1 or "internally mixed"**

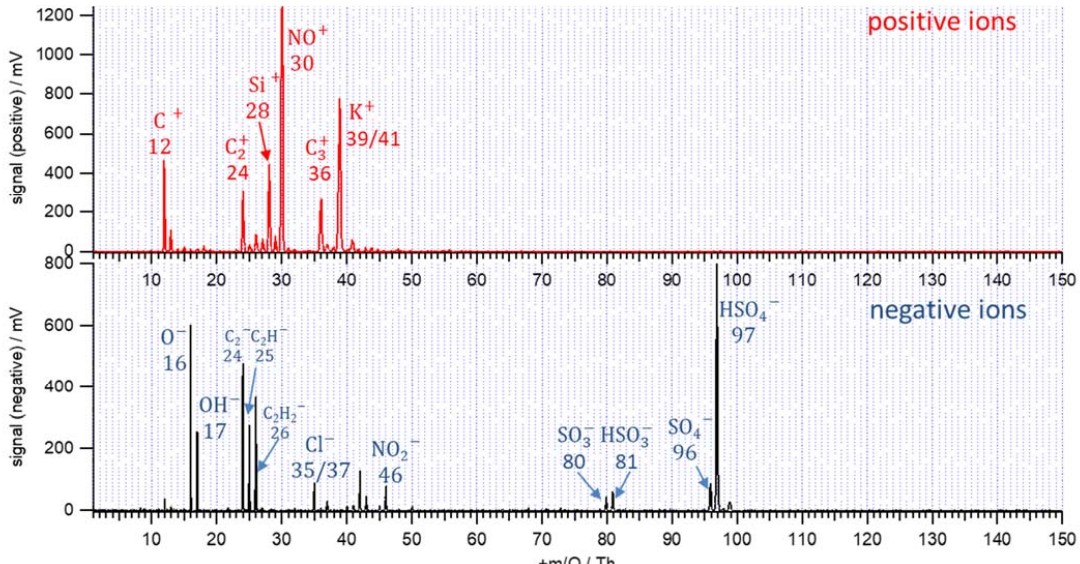

**Figure 15: Single particle spectrum corresponding to class 1 or "internally mixed"**


In summary, the ion spectra obtained from the activated ambient aerosol CCN show that the majority of the resulting droplets contain mainly the secondary aerosol components nitrates and sulphates as well as carbonaceous material. Spectra obtained for single droplets, however, show that the individual CCN can have had quite different compositions both in terms of chemical species as well as their distribution within a CCN. Some spectra show a strong signal for potassium, which indicates that at

least some of the CCN have had their origin in biomass combustion. The detection of organics in the activated CCN shows that (possibly hydrophobic) organic material did not inhibit activation of these particles.

### 5.2.1 Measurements of CCN produced from sea water samples

Activated CCN droplet spectra measured at different supersaturations are summarized in Table 3. From the total number of acquired spectra 20% to 30% were sufficiently clear for data analysis. For the initial mass calibration of the LAAPTOF (see

section 5.1) the three most dominant peaks for the positive ion spectra (i.e. $Na^+$ (m/z =23), $K^+$ potassium (m/z= 39) and $Na_2Cl^+$ (m/z=81 and 83)) were used and for the negative ions spectra the peaks for $O^-$ and $OH^-$ (m/z= 16, 17), $Cl^-$ (m/z= 35, 37), $NaCl^-$ (m/z=58) and $Na_2Cl^-$ (m/z=93 and 95).





| Sample | SS (%) | Total spectra (#) | Used spectra (#) | Spectra for analysis (%) | Positive peaks for calibration (m/z) | Negative peaks for calibration (m/z) |
|---|---|---|---|---|---|---|
| **Palma de Mallorca** | | | | | | |
| | 0.035 | 4040 | 1075 | 27% | 23/81/83 | 35/37/93 |
| | 0.054 | 4030 | 1051 | 26% | 23/39/81 | 17/35/93 |
| | 0.1 | 4040 | 1045 | 26% | 23/39/81 | 16/35/93 |
| | 0.6 | 4120 | 1585 | 38% | 23/39/81 | 17/35/37 |
| **San Sebastián** | | | | | | |
| | 0.035 | 3310 | 618 | 19% | 23/39/81 | 17/35/58 |
| | 0.054 | 4030 | 874 | 22% | 23/39/81 | 16/35/58 |
| | 0.1 | 4030 | 1010 | 25% | 23/39/81 | 17/35/58 |
| | 0.6 | 4040 | 1367 | 34% | 23/39/81 | 17/35/58 |

**Table 3: Summary of the single CCN spectra for sea water samples and strongest ion signal used for the single particle mass calibration.**

### 5.2.2 Evaluation of spectra of activated CCN produced from sea water samples

As the aerosolization of sea water samples produced internally mixed particles of homogeneous composition, only one cluster was used in the evaluation of the spectra. The resultant spectrum for each location and supersaturation is the mean of all spectra obtained for this measurement set. By comparing these mean spectra obtained at the different supersaturations, no differences in the dominant peaks are observed. Only the intensities of some peaks change slightly from one mean spectrum to the other. Results of mean spectra obtained for activated CCN (0.6% supersaturation) from sea water samples from Palma de Mallorca and San Sebastián are shown in Fig.16 -17. For samples collected at both locations, the positive ion spectra contain signs of carbon $C^+$ (m/z= 12), $Na^+$ (m/z= 23), $Mg^+$ (m/z= 24), $K^+$ (m/z= 39, 41) and $Na(H_2O)_2^+$ (m/z=59) as well as $Na_2Cl^+$ (m/z= 81,83). The peak at m/z=40 could indicate $Ca^+$ and/or $Na(H_2O)^+$. The negative ion spectra contain $O^-$ (m/z=16) and $OH^-$ (m/z=17) which correspond to water, $Na^-$ (m/z= 23), $NaCl^-$ (m/z=58, 60), $Na_2Cl^-$ (m/z=93 ,95), $MgCl_3^-$ (m/z=129,131) and $Na_2Cl_3^-$ (m/z=151,153,155).

These positive and negative peaks were also detected in the study by Prather et al. (2013), who found the positive ion peaks $Na^+$ (m/z= 23), $Mg^+$ (m/z= 24), $K^+$ (m/z= 39 and 41) and the clusters $Na_2Cl^+$ (m/z= 81 and 83) and the negative ion peaks Na- (m/z= 23) and Cl- (m/z= 35 and 37) as well as the intense alkali metal chloride clusters $NaCl^-$ (m/z=58 and 60), $Na_2Cl^-$ (m/z=93 and 95), $MgCl_3^-$ (m/z=129 and131) and $Na_2Cl_3^-$ (m/z=151,153 and155). In their study, particles with intense Na and Cl predominated in the size range above 300nm. Contrary to our study, Prather et al. (2013) did not find the $C^+$(m/z=12) peak always present in our spectra.





Shen et al. (2018a) performed mass spectroscopic analyses of single dry particles obtained from synthetic sea water. The spectra are very similar to the spectra obtained in the present study except for the peaks from carbon $C^+$ (m/z =12) and $NaCl^-$

(m/z=58, 60), which were not seen in the data sets by Shen et al. (2018a).

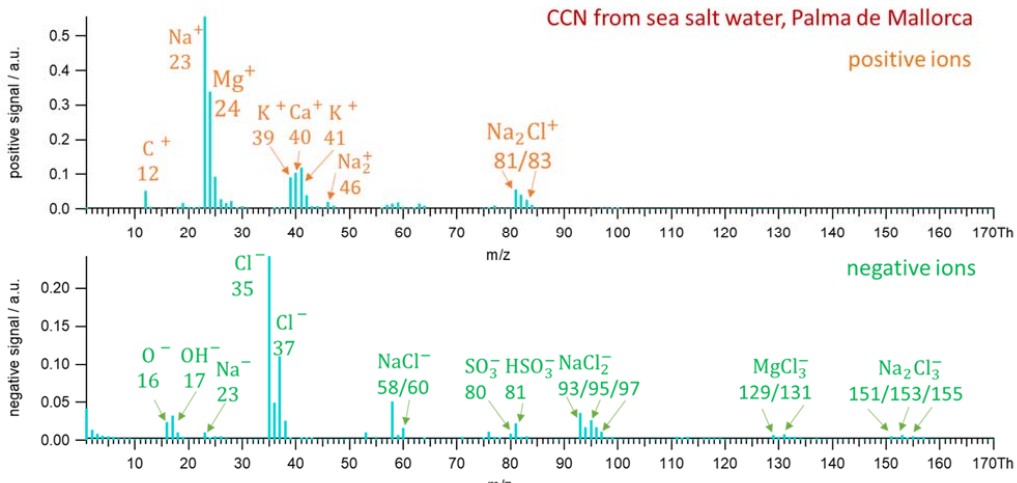

**Figure 16: Central spectrum corresponding to activated CCN (0.6% supersaturation) from sea water samples from Palma de Mallorca**

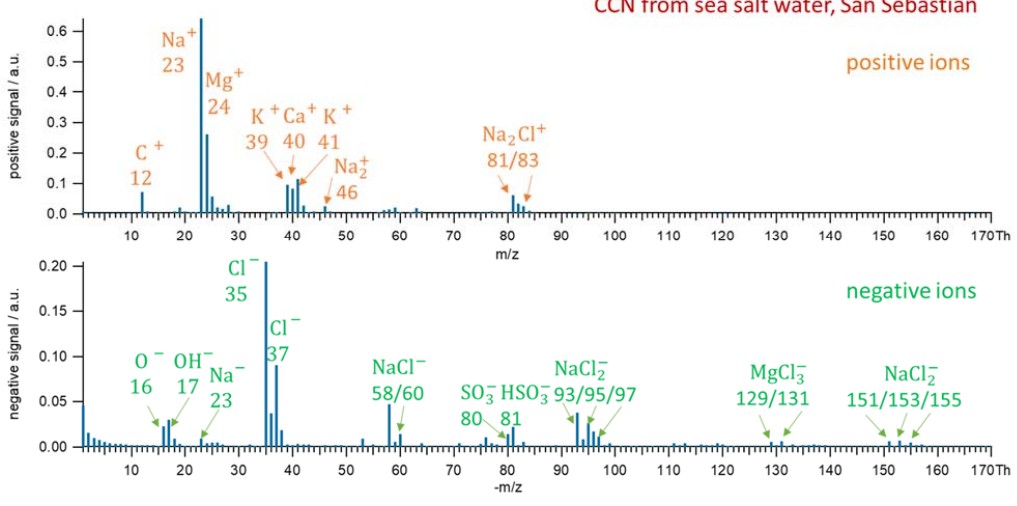





**Figure 17: Central spectrum corresponding to activated CCN (0.6% supersaturation) from sea water samples from San Sebastián**

### 5.2.3. Spectra obtained from single activated sea water CCN

As explained previously, the averaging process of the clustering method leads to a loss of information on the chemical

composition of individual particles. Figures 18 and 19 illustrate the spectrum of an activated sea water CCN (supersaturation 0.6%) from sea salt water sample from Palma de Mallorca and San Sebastián respectively. Carbon and organic peaks are observed in the negative spectrum.


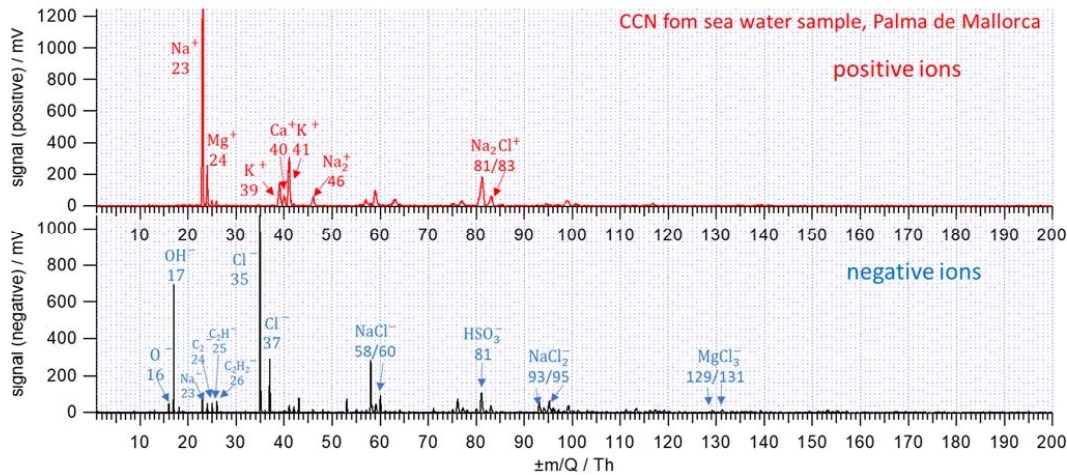

**Figure 18: Single particle spectrum corresponding to CCN particle from sea water sample, Palma de Mallorca**





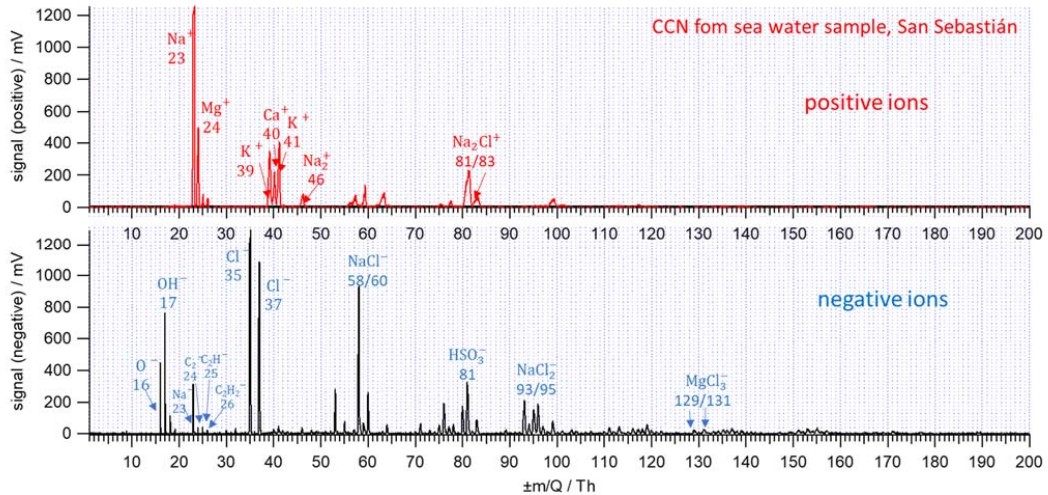

**Figure 19: Single particle spectrum corresponding to CCN particle from sea water sample, San Sebastián**

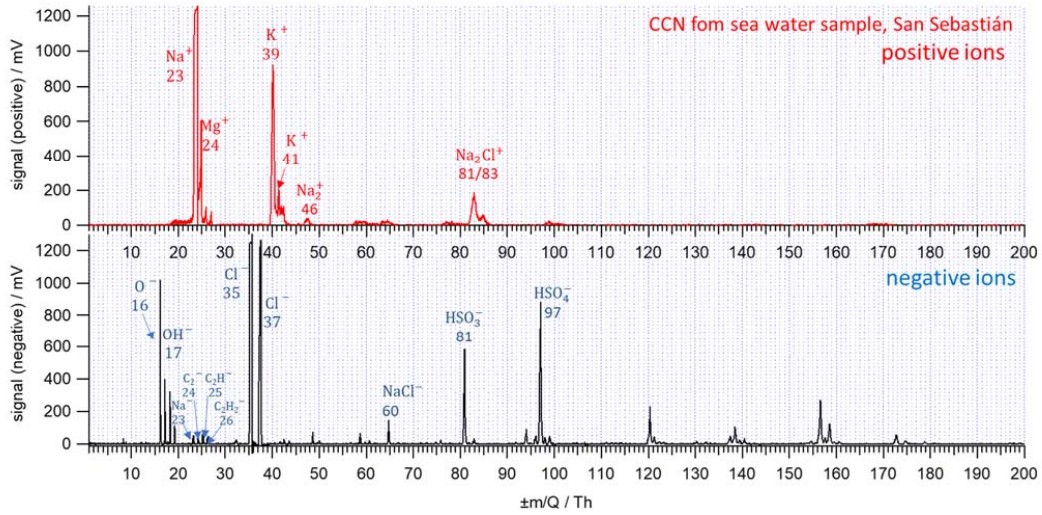

**Figure 20: Single particle spectrum corresponding to CCN particle from sea water sample, San Sebastián**






Interestingly, there are some differences in the spectra of activated CCN from the two samples. Figure (20, San Sebastián) shows more intensive peaks for the typical Cl signals. Figure (20, San Sebastián), on the other hand, shows a strong sulphate signal, which is very rarely observed in the spectra for the Palma de Mallorca sample (there is something wrong with this and the previosu sentence- they both refer to San Sebastián but read as if they correspond to different samples) . An analysis of all
spectra for all sites and supersaturations showed quite some vatiation in individual activated CCN even though the chemical composition should be the same for each sample due to the production method (aerosolization of bulk samples). The sulphate peak, e. g., was not present in all spectra, but if it was, it was quite strong in the San Sebastián CCN. In the Palma de Mallorca CCN, on the other hand, the sulphate peak was found only very rarely and, if it was, with a very weak signal.

Although the mean spectra of CCN activated at the different supersaturations produced from the two sea water samples do not
contain different major ion peaks, a comparison of the spectra obtained for individual particles shows differences in the signs of organic material and sulphate. The strong carbon signals in all spectra obtained for activated CCN indicate that the organic material present in the sea water samples from both locations did not inhibit CCN activation at the supersaturations used here.

**6 Conclusions**

Ambient aerosol particles are mostly internally mixed containing many different chemical elements and compounds (e.g. Kane
and Johnston, 2000; Hinz et al., 2005). The aim of this study was to gain information on the chemical composition of individual CCN by coupling a CCN-VACES to a LAAPTOF. The CCN-VACES provided CCN droplets activated at different supersaturations (range from 0.035 to 0.6% super saturation) with concentrations enriched by a factor of ca. 16, which were subsequently analysed on-line with the LAAPTOF to obtain both positive and negative ion spectra of individual particles. The size detection limit of the LAAPTOF was set at 350 nm, so only activated CCN were analysed. Analyses of activated CCN
droplets from ambient air were performed at five different measurement days in summer 2019. The resulting individual particle spectra mainly could be classified into three or four different classes using a fuzzy c-means cluster algorithm (Hinz et al. 1999). The membership factor $\mu$ was used to estimate the number of particles corresponding to a class. Results show no correlation of $\mu$ with supersaturation.

These classes were named according to their predominant peaks. Ammonium and sulphate were found in classes 1, 2 and 4
indicating that CCN droplets consist of secondary aerosol. Fuzzy c-means cluster algorithms provide a good overview of the average chemical composition of entire sets of spectra obtained for a specific aerosol sample, but during the process, information on individual particles is lost. The advantage of the LAAPTOF; however, is its ability to yield spectra indicating the chemical composition of single particles. Analyses of single particle spectra showed that most activated CCN indeed contained some secondary aerosol material, but there were exceptions. Some activated CCN gave strong signals of $K^+$
(potassium, m/z= 39 and 41) and organics $C_X^- H_Y^-$ (m/z= 25, 26, 27), an indication of their origin in the combustion of biomass. Some activated CCN showed highly complex spectra with a large variety of signals of organic material. Some spectra were found in the data set that closely resembled the spectra of carbon black particles used in the calibration of the time-of-flight





part of the LAAPTOF, indicating that these CCN had been probably Diesel soot. As practically all spectra contained signs for carbonaceous substances, our results show that these organics did not inhibit CCN activation.

Activated CCN originating from nebulized sea water samples collected at two different sites (Palma de Mallorca and San Sebastián, Spain) gave spectra indicating the presence of the expected major components of sea salt, but also of a large variety of organic compounds. As the CCN were internally mixed with a homogeneous composition owing to their method of production, only one cluster was selected for the spectral analysis. The averaged spectra of activated CCN from the sea water samples from the two sites are quite similar, but some individual spectra of the CCN produced from the San Sebastián sample

gave stronger signals for a larger variety of organics than those from the Palma de Mallorca sample, and some had a strong sulphate signal absent in the Palma de Mallorca CCN, showing again the value of single particle analysis.

**Author contribution**

CD: experiment performance and development; data analysis and processing; writing of MS.

AW: technical and scientific advice on LAAPTOF operation, data processing and analysis.

GS: technical and scientific advice for the experiments and the calibration of the VACES.

HS: technical support for the experiment.

CS:  developed the VACES system and provided technical support for the preparation of the manuscript.

RH: initiator and supervisor of this research work, input to experiment and data analysis, extensive input to MS text.

**Acknowledgements**

The authors wish to thank Wolfgang Ludwig, University of Vienna, for providing the fuzzy c-means clustering program used to obtain the appropriate number of clusters and Johannes Schneider, Max Planck Institute for Chemistry, Mainz, for valuable literature recommendations. Open access funding provided by University of Vienna.

**Competing interests**

The authors declare that they have no conflict of interest.

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
