# Peer review of "On-line determination of the chemical composition of single activated cloud condensation nuclei – a first investigation of single urban CCN and CCN obtained from sea water samples"

_Atmospheric Measurement Techniques, 2020_

## Referee Comment (RC1) · Anonymous Referee #1 · 24 Feb 2020

General comments:

The manuscript intended to introduce a novel method to characterize the individual activated CCN chemically. Based on a Cloud Condensation Nuclei-Versatile Aerosol Concentration Enrichment System (or CCN-VACES), particles can be firstly activated, followed by the detection. A Laser Ablation Aerosol Particle Time of Flight mass spectrometer (LAAPTOF) was deployed downstream to obtain the chemical composition of

individual CCN particles.

The experiment design and data analysis presented are quite well. However, the way that the authors presented their results might require improvement. The discussion on the results is full of details (several mass spectra of the detected particles) rather than presenting new knowledge out of such information. The authors need to provide an in-depth scientific interpretation and discussion on what is unique with this newly developed technique. Another major concern is that the major conclusions might not be supportive in the current version. More comparisons between individual activated CCN and individual droplet residues should be done to validate their results on the measurements of individual activated CCN. Barely with the mass spectral data, it is hard to confirm that the detection particles are in the form of droplets. Is it possible that the detected particles are already dried?

Overall, the topic of this manuscript is relevant to the journal and has importance scientifically. Prior to publication, the authors should also address the specific comments below.

Specific comments:

1. Introduction: the introduction of the CCN should be more specific for SSA particles. The authors just put some basic knowledge together, which is not explicitly in line with the major conclusion of this study. Some sentences, such as "Ambient aerosols originate from multiple different sources. Chemical reactions of natural and/or anthropogenic precursor gases lead to particle nucleation events.", and "Water soluble organic carbon (WSOC) has been shown to influence particle activation (e.g. McFiggans et al., 2006, Jacobson et al., 2000).", and "Cziczo et al. (2003) and Cziczo et al. (2006) coupled a continuous flow ice nuclei counter (Rogers et al., 2001) to a single particle mass spectrometer (PALMS; Murphy et al, 1998) to focus on chemical characterization of IN, and single particle analyses of ice particle residuals where conducted by Schmidt et al (2017) at Jungfraujoch." are not necessary or duplicate.

[Figure]

2. Introduction: the authors mentioned that there are challenges to measure the chemical compositions of a single ambient particle. This might be not accurate. The development of SPMS could be dated back to decades, and there are many results in this topic, as also listed in the manuscript. Besides, Aerodyne aerosol mass spectrometer (AMS) could not address this issue.

3. Introduction: while there is no study to-date focusing on the chemical composition of single activated CCN, there are probably many results on the chemical composition of cloud or ice particle residues. Rather than listing the references, I suggest that the results related to chemistry compositions of SSA should be included to make the introduction more readable. Further, an answer to why direct measurements of droplets are essential is also necessary.

4. Introduction: as noted in Line 59, "The role of the contribution of organic material to SSA in remote regions", more results on the observed of the chemical composition of single SSA in the atmosphere or cloud should be included to make it more complete.

5. Line 94 "…only a short time": please be specific.

6. Instrumentation: It is not clear enough in the text to show how to separate the particles and droplets. Even if the cut size of the virtual impactor is 1.5 micrometer, the number fraction of particles with sizes larger than this should be estimated and accounted for in such measurements. In addition, the sizes of the produced SSA should be given to evaluate the property of droplet separation.

7. Again, how to test if the separated droplets are evaporated in the vacuum before being ionized?

8. Line 275: In the negative ion spectrum signs of O (m/z=16) and OH- (m/z=17) correspond to water. References would be helpful here. Why were these peaks not shown in every droplet?

9. Figure 5-7: I do not understand why these peaks are present in the negative ions

spectrum.

10. Line 369: "The presence of Si+ (m/z = 28) in the spectrum in Fig.12 might indicate a CCN not originating in the area of Vienna.". Such a statement is not satisfying and does not help in the discussion.

11. Section 5.2.2: Is there only one type of SSA? This is apparently different from previous laboratory studies.

12. Section 5.2.3: I would recommend the authors focus more on what is new about this mass spectra. Does it provide more information than those for droplet residues? Only with more validation, can the author state that such a technique would provide insights into the composition of individual activated CCN.

---

## Referee Comment (RC2) · Anonymous Referee #2 · 28 Mar 2020

Dameto de España et al present results of coupling a CCN-VACES to a LAAPTOF in gain information about the chemical composition of individual CCN. I believe this is a worthwhile endeavor to pursue (single-particle mass spectrometry of previously activated particles); however, I agree with the comments of Reviewer 1 about the authors' claims about their data, especially in considering their experimental design, data checks, and data interpretation with respect to their stated goal and asserted conclusions. Further experimental and data analysis work are necessary to improve the

quality of the work and support the claims made. The analysis of the chemical composition of activated urban CCN and SSA are worthwhile endeavors and likely deserve separate papers themselves focused on the details of the science results obtained, as a detailed investigation of the science is lacking here, with only a cursory examination of the mass spectra is provided. My detailed comments are provided below.

In reviewing the references, I have two major concerns. There are 4 references included that are non-peer-reviewed conference abstracts (including an EGU abstract, for example). In the main text, these citations seem to suggest more confidence in the references than is warranted. For example, Lines 167-168 states "A first pilot study on the ability of the LAAPTOF to detect and analyse aqueous droplets was performed by Dameto de España et al. (2018b)." This is a reference to the 2018 Aerosol Technology Conference in Bilbao, Spain; I checked the conference website and could not even find an abstract to refer to, only the presentation title. In addition, the reference list includes four references without journals or DOIs provided, so they could not be traced. Shen et al 2018b should be updated from the AMTD to AMT version.

As Reviewer 1 also discusses, the authors assert that the LAAPTOF measured aqueous droplets (as a main conclusion of their paper), but no information about particle size is provided in the data presented, and this is needed to confirm their hypothesis. To test this assertion, it would be best measure the particle size following the CCN instrument and within the LAAPTOF for comparison. What is the size distribution of activated droplets? What is the size distribution measured by the LAAPTOF? Without knowledge of the particle size, it is not possible to determine whether the particles analyzed were indeed aqueous droplets when measured. Since the impactor removed particles greater than 1.5 um, then any particle detected by the LAAPTOF at less than 1.5 um had lost water. Even particles measured by the LAAPTOF at greater than 1.5 um may have also partially evaporated if they had started out as much larger droplets. Also key to the assertion of the measurement of aqueous droplets is the work of Zelenyuk et al (2006, Analytical Chemistry, "Evaporation of water from particles in an

aerodynamic lens inlet: An experimental study"), which was completed using a single-particle mass spectrometer, and is not cited by the authors.

I agree with Reviewer 1 that the introduction needs revision. The general components of the introduction would be best referenced using reviews to more comprehensively cover the literature and avoid inaccurate statements. The authors should also focus on previous work that can be most directly compared to this study. For example, on lines 61-62, the authors discuss SSA smaller than 200 nm (even though this study only examined SSA > 300 nm) with two example papers, one of which didn't even measure SSA (Pratt et al 2009). Another inaccurate statement is on lines 71-72 where it is stated that "single particle laser ablation. . .been widely used to analyse bulk chemical composition". The statement on Line 90 "All these studies performed with the various types of single particle mass spectrometer analysed dry particles." is also inaccurate, as driers are not always used upstream of single particle mass spectrometers; the authors should better review the literature and then amend this statement, as driers are often NOT used. It is also inaccurately stated that Neubauer et al (1997) analyzed "aqueous solutions", when in fact they analyzed aqueous aerosol. Also, lines 98-104 do not present a thorough review of single-particle mass spectrometry measurements of particles <0.2 um in diameter, as many other studies have examined this size range; however, this study doesn't examine this size range either so reviewing these papers doesn't seem necessary. It is important to instead include more discussion of previous single-particle mass spectrometry measurements of cloud droplet residuals, as this is most relevant to this work, and is key for establishing the uniqueness of this work.

Line 157: The authors state that particles form 70 nm – 2.5 um are transmitted with 100% efficiency and cite the instrument user manual, but I am not aware of any published study proving this. Further, the authors themselves note that spectra were only obtained for particles > 0.3 um, and they do not provide data showing the transmission efficiency of their instrument as setup for this study. I am not aware of any aerodynamic lens inlet available that transmits this full particle size range at 100% efficiency.

The authors refer to Liu et al (1995a, 1995b) for their aerodynamic lens, but Liu et al (1995b), which shows experimental data, does not show 100% transmission over this size range either.

Lines 162-164: The goal of this work is to measure aqueous droplets >1.5 um in diameter, but the authors only do a PSL size calibration from 350-1500 nm, meaning that, if they did measure larger particles (diameter not reported), they would be extrapolating their size calibration and significantly increasing uncertainty. To claim that the droplets did not lose water, the particle size is critical to report, and as such, the authors should increase their size calibration to the full 2.5 um, which they state is their upper size range.

Figures 1 & 3: These figures show the LAAPTOF diagram and carbon black mass calibration spectra, but since this paper does not focus on the development of the LAAPTOF itself, these figures are not needed. As listed in Section 2.2, there are already several papers published on the LAAPTOF.

Section 5.1: This section primarily discusses the mass calibration and spectra clustering setup. This information should be moved to the methods, as this is not new to this work, with several papers published already on these topics. Likewise, the columns in Tables 2 & 3 that correspond to the peaks chosen for calibration should not be in the results.

Lines 216-217: It is stated that "ca. 10% to 20% of the spectra obtained from the droplets were sufficiently clear for analysis". What does this mean? I'm concerned that only 10-20% of the data obtained were analyzed, per Table 2, as this could bias the results. Since water suppresses negative ion formation (Neubauer et al, 1997), does this mean that only 10-20% of the particles dried to the point that mass spectra could be obtained? What fraction of the particles had negative ions? In fact, this information this could potentially help support the claims of water still present on the particles. How does the measured size distribution of these particles compare to those without

"sufficiently clear" mass spectra?

Line 269: The authors claim here and elsewhere that m/z 18 is NH4+ and cite Brands et al (2011) and Hinz et al (2006). However, these instruments use higher wavelength lasers for desorption/ionization (Brands et al uses 266 nm and Hinz et al used 266 nm and 337 nm), at which m/z 18 does indeed correspond to NH4+. However, these authors are using 193 nm, and as discussed by Murphy and Thomson (1997a, JGR), m/z 18 can correspond to both NH4+ and H2O+ (water) at this wavelength; this may even be helpful for the authors to examine further since they are studying aqueous aerosol.

Lines 275: The authors claim here and elsewhere that m/z -16 and -17 (O- and OH-) correspond to water, but no reference is provided. Murphy and Thomson (1997b, JGR), who also used 193 nm LDI, found that O- and OH- were negatively correlated with relative humidity and were positively correlated with organic and silicon peaks instead.

Lines 276-277: The authors refer to m/z -46 & -62 (NO2- and NO3-) and m/z -80 and -97 (SO3- and HSO4-) as ammonium nitrate and ammonium sulfate here and elsewhere in the manuscript. However, it is not clear how the paired cation was determined, especially since not all particle types included ammonium.

Figure 11: Why is this particle labeled "carbon and nitrate" when one of the most prominent peaks is sulfate?

Lines 441-444: The first two lines refer to a comparison of two samples, but seems to refer to Figure 20 twice, making this not understandable. It is then followed by perhaps a statement left in from coauthor editing(?) that states: "there is something wrong with this and the previosu sentence-they both refer to San Sebastian but read as if they correspond to different samples". This should be fixed.

Lines 460-463: Information about particle clustering is not novel to this paper and

therefore does not belong in the conclusions.

---

## Author Comment (AC1) · 25 May 2020

***The authors' comments to the individual points raised by the reviewer are given below in italics***

General comments:

The manuscript intended to introduce a novel method to characterize the individual activated CCN chemically. Based on a Cloud Condensation Nuclei-Versatile Aerosol Concentration Enrichment System (or CCN-VACES), particles can be firstly activated, followed by the detection. A Laser Ablation Aerosol Particle Time of Flight mass spectrometer (LAAPTOF) was deployed downstream to obtain the chemical composition of

individual CCN particles.

The experiment design and data analysis presented are quite well. However, the way that the authors presented their results might require improvement. The discussion on the results is full of details (several mass spectra of the detected particles) rather than presenting new knowledge out of such information. The authors need to provide an in-depth scientific interpretation and discussion on what is unique with this newly developed technique. Another major concern is that the major conclusions might not be supportive in the current version. More comparisons between individual activated CCN and individual droplet residues should be done to validate their results on the measurements of individual activated CCN. Barely with the mass spectral data, it is hard to confirm that the detection particles are in the form of droplets. Is it possible that the detected particles are already dried?

*The main question raised by the reviewer is: what is new? The answer to this question is: with this setup, we show that individual CCN can be separated from the general aerosol population and analysed individually, rather than aerosol particles in general. In short, the CCN-VACES activates CCN at a specific (selected) supersaturation, separates them from non-activated particles and enriches their concentration in the virtual impactor. These activated CCN are transferred as droplets into the LAAP-TOF and analysed there. Single particle mass spectrometry cannot determine whether a particle is a CCN, but with the link to the CCN-VACES, "true" CCN can be analysed. To our knowledge, this has not been done before for previously non-activated CCN.*

*A point already made here: the LAAP-TOF measures what is transferred to it, so the question of comparing spectra of droplets and droplet residues is not pertinent in our study – even if the droplets had dried to some extent by the time they passed the sizing laser triggering the ablation laser (in our case the lower size limit was set to 300nm), the particles/droplets that are ablated to obtain spectra are particles/droplets that had been actually activated CCN, but not true cloud droplet residues.*

*We do see the point of this reviewer, however, will remove details and will shorten and re-focus the MS to clarify this main point.*

Overall, the topic of this manuscript is relevant to the journal and has importance scientifically. Prior to publication, the authors should also address the specific comments below.

Specific comments:

1. Introduction: the introduction of the CCN should be more specific for SSA particles. The authors just put some basic knowledge together, which is not explicitly in line with the major conclusion of this study. Some sentences, such as "Ambient aerosols originate from multiple different sources. Chemical reactions of natural and/or anthropogenic precursor gases lead to particle nucleation events.", and "Water soluble organic carbon (WSOC) has been shown to influence particle activation (e.g. McFiggans et al., 2006, Jacobson et al., 2000).", and "Cziczo et al. (2003) and Cziczo et al. (2006) coupled a continuous flow ice nuclei counter (Rogers et al., 2001) to a single particle mass spectrometer (PALMS; Murphy et al, 1998) to focus on chemical characterization of IN, and single particle analyses of ice particle residuals where conducted by Schmidt et al (2017) at Jungfraujoch." are not necessary or duplicate.

*The introduction will be considerably shortened and focused on the main topic of the paper. SSA and the literature studies of cloud droplet residues (see reviewer comment #3) will be given more attention, and the difference between CCN and droplet residues will be clarified.*

2. Introduction: the authors mentioned that there are challenges to measure the chemical compositions of a single ambient particle. This might be not accurate. The development of SPMS could be dated back to decades, and there are many results in this topic, as also listed in the manuscript. Besides, Aerodyne aerosol mass spectrometer (AMS) could not address this issue.

*We do not deal with aerosol particles in general (and we definitely do and did not say that the Aerodyne instrument can measure single particles), but with a specific subset of the aerosol, i.e. CCN, which are defined as those particles, which can be activated and grow to (large) droplets under specific supersaturated conditions. The problem is not how to analyse single particles (this can be done by a number of already existing instruments), but to extract the CCN out of the total set of particles and analyse only these CCN. This point will be clarified in the revised MS*

3. Introduction: while there is no study to-date focusing on the chemical composition of single activated CCN, there are probably many results on the chemical composition of cloud or ice particle residues. Rather than listing the references, I suggest that the results related to chemistry compositions of SSA should be included to make the introduction more readable.

*Yes, there are several studies focusing on the chemical composition of cloud droplet or ice particles residues. We did not previously include these studies in our MS, as we focus on CCN. The scientific question behind our experimental setup is: which particles can be activated and what is their chemical composition. Cloud droplet residues may have undergone chemical reactions during their "life" as cloud droplets, while the CCN that can be studied with our setup are those particles that can form cloud droplets after activation.*

Further, an answer to why direct measurements of droplets are essential is also necessary.

*The focus was on activated CCN, which are of course droplets. Had these droplets been dried to their original size, they would have been too small to be analysed with the LAAP-ToF, which was set intentionally to analyse only particles >300nm to exclude not activated (i.e. non-CCN) particles still present in the minor flow of the virtual impactor.*

4. Introduction: as noted in Line 59, "The role of the contribution of organic material to SSA in remote regions", more results on the observed of the chemical composition of single SSA in the atmosphere or cloud should be included to make it more complete.

*This info will be added. Additionally, we will include already in the introduction that we do not analyse SSA, but nebulized (re-aerosolized) sea water samples.*

5. Line 94 "…only a short time": please be specific.

*The time probably is of the order of less than 1 second. As the LAAP ToF manual is not specific in this respect, we cannot give a more precise estimate.*

*The activated CCN enter the LAAP-ToF definitely as droplets, as the water vapour content of the aerosol stream is definitely above the efflorescence humidities (typically around 40%).*

6. Instrumentation: It is not clear enough in the text to show how to separate the particles and droplets. Even if the cut size of the virtual impactor is 1.5 micrometer, the number fraction of particles with sizes larger than this should be estimated and accounted for in such measurements. In addition, the sizes of the produced SSA should be given to evaluate the property of droplet separation.

*For atmospheric particles, the number size distribution drops sharply towards the large size end. In our large data base on number size distributions measured in Vienna, concentrations of particles >1.5 µm are typically at least four orders of magnitude less than the concentrations of particles below this size. Activation diameters even of insoluble particles at the supersaturations we use are well below 1 µm, so the probability of having unactivated particles with sizes >1.5 µm is negligible.*

*To be precise: we do not measure SSA, but we nebulize (re-aerosolize) sea water samples in a Collison Atomizer, so we do not have real SSA. The droplets coming from the Collison atomizer are dried and then drawn into the CCN-VACES for activation. All droplets with sizes >1.5µm resulting from the activation of these dry particles are then passed to the LAAP-ToF.*

7. Again, how to test if the separated droplets are evaporated in the vacuum before being ionized?

*Because of the construction of the LAAP-ToF, such a test can unfortunately not be performed*

8. Line 275: In the negative ion spectrum signs of O (m/z=16) and OH- (m/z=17) correspond to water. References would be helpful here. Why were these peaks not shown in every droplet?

*The comment here refers to the central spectra obtained from fuzzy clustering. Fuzzy clustering suppresses weak signals, and as these signals are rather weak, they are not visible in these spectra.*

9. Figure 5-7: I do not understand why these peaks are present in the negative ions

spectrum.

*Does this refer to the O and OH peaks?*

10. Line 369: "The presence of Si+ (m/z = 28) in the spectrum in Fig.12 might indicate a CCN not originating in the area of Vienna.". Such a statement is not satisfying and does not help in the discussion.

*The statement will be deleted. The main point here is that we saw fine Si-rich particles mainly in long-range transport air masses (Okada et al, 2001)*

11. Section 5.2.2: Is there only one type of SSA? This is apparently different from previous laboratory studies.

    *We analyse CCN originating from two different sea water samples – as stated above, we will state already in the introduction that we do not analyse real world SSA*

12. Section 5.2.3: I would recommend the authors focus more on what is new about this mass spectra. Does it provide more information than those for droplet residues? Only with more validation, can the author state that such a technique would provide insights into the composition of individual activated CCN.

*There is a difference between droplet residues and CCN before activation. Droplet residues have spent time in clouds and their chemical composition might have changed because of chemical reactions and / or coalescence with other cloud droplets. We look at the initial state of activated CCN . This point will be clarified in the revised MS*

---

## Author Comment (AC2) · 25 May 2020

***The authors' comments to the individual points raised by the reviewer are given below in italics***

Dameto de España et al present results of coupling a CCN-VACES to a LAAPTOF in gain information about the chemical composition of individual CCN. I believe this is a worthwhile endeavor to pursue (single-particle mass spectrometry of previously activated particles); however, I agree with the comments of Reviewer 1 about the authors' claims about their data, especially in considering their experimental design, data checks, and data interpretation with respect to their stated goal and asserted conclusions.

*See our comments to the points raised by reviewer 1*

Further experimental and data analysis work are necessary to improve the quality of the work and support the claims made. The analysis of the chemical composition of activated urban CCN and SSA are worthwhile endeavours and likely deserve separate papers themselves focused on the details of the science results obtained, as a detailed investigation of the science is lacking here, with only a cursory examination of the mass spectra is provided. My detailed comments are provided below.

*The current MS is a technical paper to demonstrate the feasibility of our main aim, i.e. that the separation of activated CCN and their chemical analysis can be done, and not a paper on measurement campaigns. The challenge is: **a)** to separate CCN (i.e. particles that can be activated to grow to large droplets under conditions of specific supersaturations) from the non-CCN aerosol, and: **b)** to analyse the chemical composition of the individual CCN. Real world CCN are a subfraction of the atmospheric aerosol and – except for their activation behaviour – not distinguished by unique characteristics that can be easily measured. In the urban aerosol of Vienna we see that the measured activation ratio of the aerosol is much lower than the activation ratio expected from the size distribution (Burkart et al, 2012, Dameto de Espana et al., 2017), so size is not the main characteristic.*

In reviewing the references, I have two major concerns. There are 4 references included that are non-peer-reviewed conference abstracts (including an EGU abstract, for example). In the main text, these citations seem to suggest more confidence in the references than is warranted. For example, Lines 167-168 states "A first pilot study on the ability of the LAAPTOF to detect and analyse aqueous droplets was performed by Dameto de España et al. (2018b)." This is a reference to the 2018 Aerosol Technology Conference in Bilbao, Spain; I checked the conference website and could not even find an abstract to refer to, only the presentation title. In addition, the reference list includes four references without journals or DOIs provided, so they could not be traced. Shen et al 2018b should be updated from the AMTD to AMT version.

*The references to the earlier conferences were added to show that parts of the study were already presented at conferences, so we think that these references should be included. We will, however, clarify already in the text that these are conference presentations and not full-scale peer reviewed papers.*

*The reference Shen et al. will of course be updated*

As Reviewer 1 also discusses, the authors assert that the LAAPTOF measured aqueous droplets (as a main conclusion of their paper), but no information about particle size is provided in the data presented, and this is needed to confirm their hypothesis. To test this assertion, it would be best measure the particle size following the CCN instrument and within the LAAPTOF for comparison.

*Because of the set-up of the LAAP-ToF (a closed instrument that cannot be taken apart) this measurement is not possible*

What is the size distribution of activated droplets?

*The size distribution of droplets produced by the original VACES has been investigated previously e. g. by Saarikoski S., Carbone S, Cubison MJ, Hillamo R, Keronen P., Sioutas C., Worsnop D.R, Jimenez J.L." Evaluation of the performance of a particle concentrator for on-line instrumentation". Atmospheric Measurement Techniques 7, 2121-2135, 2014; and Geller, M.D., Biswas, S., Fine, P.M. and Sioutas, C. "A Compact Aerosol Concentrator for Use in Conjunction With Low Flow-Rate Continuous Aerosol Instrumentation." Journal of Aerosol Science,36, 1006-1022, 2005. The size distribution of the droplets in these studies had it maximum between 4 and 5 µm. This info will be added to the revised MS.*

*The main point for our MS, however, is that the droplets are larger than the cut size of the virtual impactor of the VACES (1.5 µm) and, when they arrive at the point of ablation, larger than 300nm (the size limit set in our experiment for the triggering of the ablation laser).*

What is the size distribution measured by the LAAPTOF? Without knowledge of the particle size, it is not possible to determine whether the particles analyzed were indeed aqueous droplets when measured. Since the impactor removed particles greater than 1.5 um, then any particle detected by the LAAPTOF at less than 1.5 um had lost water. Even particles measured by the LAAPTOF at greater than 1.5 um may have also partially evaporated if they had started out as much larger droplets.

*All activated CCN (i.e. droplets) exiting the virtual impactor of the VACES are >1.5 µm at that point. The LAAP-ToF is intentionally set to 300 nm lower size limit to analyse also droplets that had already evaporated to some extent.*

Also key to the assertion of the measurement of aqueous droplets is the work of Zelenyuk et al (2006, Analytical Chemistry, "Evaporation of water from particles in an

aerodynamic lens inlet: An experimental study"), which was completed using a single particle mass spectrometer, and is not cited by the authors.

*The reference will be added and discussed. Zelenyuk et al. however, dealt with particles that had been grown under sub-saturated conditions (maximum relative humidity: 85%), where droplets attain a stable size. In our experiments we deal with CCN activated at supersaturated conditions, where the droplets no longer are in equilibrium and do not have stable (wet) sizes*

I agree with Reviewer 1 that the introduction needs revision. The general components of the introduction would be best referenced using reviews to more comprehensively cover the literature and avoid inaccurate statements. The authors should also focus on previous work that can be most directly compared to this study. For example, on lines 61-62, the authors discuss SSA smaller than 200 nm (even though this study only examined SSA > 300 nm) with two example papers, one of which didn't even measure SSA (Pratt et al 2009).

*See our comments to the points raised by Reviewer #1 – the introduction will be revised. The difference between SSA and our sea water CCN will be clarified already in the introduction – we do not analyse real world SSA.*

Another inaccurate statement is on lines 71-72 where it is stated that "single particle laser ablation…been widely used to analyse bulk chemical composition". The statement on Line 90 "All these studies performed with the various types of single particle mass spectrometer analysed dry particles." is also inaccurate, as driers are not always used upstream of single particle mass spectrometers; the authors should better review the literature and then amend this statement, as driers are often NOT used.

*The introduction will be revised and any inaccurate statements will be corrected.*

It is also inaccurately stated that Neubauer et al (1997) analyzed "aqueous solutions", when in fact they analyzed aqueous aerosol. Also, lines 98-104 do not present a thorough review of single-particle mass spectrometry measurements of particles <0.2 um in diameter, as many other studies have examined this size range; however, this study doesn't examine this size range either so reviewing these papers doesn't seem necessary. It is important to instead include more discussion of previous single-particle mass spectrometry measurements of cloud droplet residuals, as this is most relevant to this work, and is key for establishing the uniqueness of this work.

*Studies on cloud droplet residuals will be discussed. The main difference is that we analyse the composition of particles that can be activated as CCN, while cloud droplet residues are particles that have potentially undergone chemical reactions and/ or coalescence with other droplets during their "life" as cloud droplets*

Line 157: The authors state that particles form 70 nm – 2.5 um are transmitted with 100% efficiency and cite the instrument user manual, but I am not aware of any published study proving this. Further, the authors themselves note that spectra were only obtained for particles > 0.3 um, and they do not provide data showing the transmission efficiency of their instrument as setup for this study. I am not aware of any aerodynamic lens inlet available that transmits this full particle size range at 100% efficiency.

*Here we have to rely on the user manual. We intentionally set the lower size detection limit at 0.3 µm to exclude smaller particles.*

The authors refer to Liu et al (1995a, 1995b) for their aerodynamic lens, but Liu et al (1995b), which shows experimental data, does not show 100% transmission over this size range either.

*Again, we have to rely on the user manual*

Lines 162-164: The goal of this work is to measure aqueous droplets >1.5 um in diameter, but the authors only do a PSL size calibration from 350-1500 nm, meaning that, if they did measure larger particles (diameter not reported), they would be extrapolating their size calibration and significantly increasing uncertainty. To claim that the droplets did not lose water, the particle size is critical to report, and as such, the authors should increase their size calibration to the full 2.5 um, which they state is their upper size range.

*We do not claim that the droplets do not lose water between the exit of the CCN-VACES and the ablation stage – we claim that droplets enter the LAAP-ToF and can be analysed. We do not make any statement as to size (except that they are >0.3µm when the ablation laser is triggered). The whole MS will be checked and this point will be clarified where needed.*

Figures 1 & 3: These figures show the LAAPTOF diagram and carbon black mass calibration spectra, but since this paper does not focus on the development of the LAAPTOF itself, these figures are not needed. As listed in Section 2.2, there are already several papers published on the LAAPTOF.

*We have to write about the mass calibration we performed. As this is a small part of the MS, we prefer to keep this section*

Section 5.1: This section primarily discusses the mass calibration and spectra clustering setup. This information should be moved to the methods, as this is not new to this work, with several papers published already on these topics. Likewise, the columns in Tables 2 & 3 that correspond to the peaks chosen for calibration should not be in the results.

*This will be done in the revised MS*

Lines 216-217: It is stated that "ca. 10% to 20% of the spectra obtained from the droplets were sufficiently clear for analysis". What does this mean? I'm concerned that only 10-20% of the data obtained were analyzed, per Table 2, as this could bias the results.

*We can only work with clear spectra, and can only work with what we have. The main message of the MS is that we can obtain spectra of _single_ activated CCN. The clusters are shown to give a general overview of the CCN population which gave us spectra. We do not make a general statement on all CCN*

Since water suppresses negative ion formation (Neubauer et al, 1997), does this mean that only 10-20% of the particles dried to the point that mass spectra could be obtained? What fraction of the particles had negative ions? In fact, this information this could potentially help support the claims of water still present on the particles

*Unfortunately, the LAAP-ToF software cannot provide the number of "positive only" spectra, so we do not have the info necessary to estimate the percentage of particles without negative spectra.*

.

How does the measured size distribution of these particles compare to those without

"sufficiently clear" mass spectra?

*Unfortunately this cannot be done – we get the spectra from the LAAP-ToF, but not the size distribution of the particles/droplets.*

Line 269: The authors claim here and elsewhere that m/z 18 is $NH_4^+$ and cite Brands et al (2011) and Hinz et al (2006). However, these instruments use higher wavelength lasers for desorption/ionization (Brands et al uses 266 nm and Hinz et al used 266 nm and 337 nm), at which m/z 18 does indeed correspond to $NH_4^+$. However, these authors are using 193 nm, and as discussed by Murphy and Thomson (1997a, JGR), m/z 18 can correspond to both $NH_4^+$ and $H_2O^+$ (water) at this wavelength; this may even be helpful for the authors to examine further since they are studying aqueous aerosol.

*Thank you for this helpful suggestion. After going back to the spectra, we found, however, that in most cases the ions usually associated with NH4+ (i.e. NO3- or SO42-) were present, so we deduce that in most cases  m/z 18 indeed must have been NH4+*

Lines 275: The authors claim here and elsewhere that m/z -16 and -17 (O- and OH-) correspond to water, but no reference is provided. Murphy and Thomson (1997b, JGR), who also used 193 nm LDI, found that O- and OH- were negatively correlated with relative humidity and were positively correlated with organic and silicon peaks instead.

*The references for assigning O- and OH- to water are the studies by Johnston (2000) and Neubauer et al. (1997).  We will check for a possible correlation between O- and OH- and organic peaks as suggested.*

Lines 276-277: The authors refer to m/z -46 & -62 (NO2- and NO3-) and m/z -80 and 97 (SO3- and HSO4-) as ammonium nitrate and ammonium sulfate here and elsewhere in the manuscript. However, it is not clear how the paired cation was determined, especially since not all particle types included ammonium.

*The comment here refers to the central spectra obtained from fuzzy clustering. Fuzzy clustering suppresses weak signals, and as e.g. Na+  signals are rather weak, they are not visible in these spectra.*

Figure 11: Why is this particle labeled "carbon and nitrate" when one of the most prominent peaks is sulfate?

*This mistake will be fixed*

Lines 441-444: The first two lines refer to a comparison of two samples, but seems to refer to Figure 20 twice, making this not understandable. It is then followed by perhaps a statement left in from coauthor editing(?) that states: "there is something wrong with this and the previosu sentence-they both refer to San Sebastian but read as if they correspond to different samples". This should be fixed.

*This will be fixed*

Lines 460-463: Information about particle clustering is not novel to this paper and

therefore does not belong in the conclusions.

*We agree – this info will be deleted*

---

## Author Comment (AC3) · 25 Sep 2020

***The authors' comments to the individual points raised by the reviewer are given below in italics and in blue***

General comments:

The manuscript intended to introduce a novel method to characterize the individual activated CCN chemically. Based on a Cloud Condensation Nuclei-Versatile Aerosol Concentration Enrichment System (or CCN-VACES), particles can be firstly activated, followed by the detection. A Laser Ablation Aerosol Particle Time of Flight mass spectrometer (LAAPTOF) was deployed downstream to obtain the chemical composition of individual CCN particles.

The experiment design and data analysis presented are quite well. However, the way that the authors presented their results might require improvement. The discussion on the results is full of details (several mass spectra of the detected particles) rather than presenting new knowledge out of such information. The authors need to provide an in-depth scientific interpretation and discussion on what is unique with this newly developed technique. Another major concern is that the major conclusions might not be supportive in the current version. More comparisons between individual activated CCN and individual droplet residues should be done to validate their results on the measurements of individual activated CCN. Barely with the mass spectral data, it is hard to confirm that the detection particles are in the form of droplets. Is it possible that the detected particles are already dried?

*The main question raised by the reviewer is: what is new? The answer to this question is: with this setup, we show that individual CCN can be analysed rather than aerosol particles in general. In short, the CCN-VACES activates CCN at a specific (selected) supersaturation, separates them from non-activated particles and enriches them in concentration in the virtual impactor. These activated CCN are transferred as droplets into the LAAPTOF and analysed there. Single particle mass spectrometry cannot determine whether a particle is a CCN, and with the link to the CCN-VACES, "true" CCN can be analysed. To our knowledge, this has not been done before for previously non activated CCN.*

*A point already made here: the LAAPTOF measures what is transferred to it, so the question of comparing spectra of droplets and droplet residues is not pertinent in our study – even if the droplets had dried to some extent by the time they passed the sizing laser triggering the ablation laser (in our case the lower size limit was set to 300 nm), the particles/droplets that are ablated to obtain spectra are particles/droplets that had been actually activated CCN, but not true cloud droplet residues.*

*We do see the point of this reviewer, however, will remove details and will shorten and re-focus the MS to clarify this main point.*

Overall, the topic of this manuscript is relevant to the journal and has importance scientifically. Prior to publication, the authors should also address the specific comments below.

Specific comments:

1. Introduction: the introduction of the CCN should be more specific for SSA particles. The authors just put some basic knowledge together, which is not explicitly in line with the major conclusion of this study. Some sentences, such as "Ambient aerosols originate from multiple different sources. Chemical reactions of natural and/or anthropogenic precursor gases lead to particle nucleation events.", and "Water soluble organic carbon (WSOC) has been shown to influence particle activation (e.g. McFiggans et al., 2006, Jacobson et al., 2000).", and "Cziczo et al. (2003) and Cziczo et al. (2006) coupled a continuous flow ice nuclei counter (Rogers et al., 2001) to a single particle mass spectrometer (PALMS; Murphy et al, 1998) to focus on chemical characterization of IN, and single particle analyses of ice particle residuals where conducted by Schmidt et al (2017) at Jungfraujoch." are not necessary or duplicate.

   *The introduction will be considerably shortened and focused on the main topic of the paper. SSA and the literature studies of cloud droplet residues (see reviewer comment #3) will be given more attention.* **done**

2. Introduction: the authors mentioned that there are challenges to measure the chemical compositions of a single ambient particle. This might be not accurate. The development of SPMS could be dated back to decades, and there are many results in this topic, as also listed in the manuscript. Besides, Aerodyne aerosol mass spectrometer (AMS) could not address this issue.

   *We do not deal with aerosol particles in general (and we definitely do not say that the Aerodyne instrument can measure single particles), but with a* *specific subset of the aerosol, i.e. CCN, which are defined as those particles, which can be activated and grow to (large) droplets under specific supersaturated conditions**. The problem is not how to analyse single particles (this can be done by a number of already existing instruments), but to extract the CCN out of the total set of particles and analyse only these CCN. This point will be clarified in the revised MS*

3. Introduction: while there is no study to-date focusing on the chemical composition of single activated CCN, there are probably many results on the chemical composition of cloud or ice particle residues. Rather than listing the references, I suggest that the results related to chemistry compositions of SSA should be included to make the introduction more readable.

   *Yes, there are several studies focusing on the chemical composition of cloud droplet or ice particles residues. We did not include these studies in our MS, as we focus on CCN. The scientific question behind our experimental setup is: which particles can be activated and what is their chemical composition. Cloud droplet residues may have undergone chemical reactions during their "life" as cloud droplets, while the CCN that can be studied with our setup are those particles that can form cloud droplets after activation.*

   Further, an answer to why direct measurements of droplets are essential is also necessary.

   *The focus was on activated CCN, which are of course droplets. Had these droplets been dried to their original size, they would have been too small to be analysed with the LAAPTOF, which was set intentionally to analyse only particles >300 nm to exclude unactivated (i.e. non-CCN) particles still present in the minor flow of the virtual impactor.*

4. Introduction: as noted in Line 59, "The role of the contribution of organic material to SSA in remote regions", more results on the observed of the chemical composition of single SSA in the atmosphere or cloud should be included to make it more complete.

*It will be added. Additionally, we will include already in the introduction that we do not analyse SSA, but nebulized (re-aerosolized) sea water samples.*

5. Line 94 "…only a short time": please be specific.

*The activated CCN enter the LAAPTOF definitely as droplets, as the water vapour content of the aerosol stream is definitely above the efflorescence humidities (typically around 40%).*

6. Instrumentation: It is not clear enough in the text to show how to separate the particles and droplets. Even if the cut size of the virtual impactor is 1.5 micrometer, the number fraction of particles with sizes larger than this should be estimated and accounted for in such measurements. In addition, the sizes of the produced SSA should be given to evaluate the property of droplet separation.

*For atmospheric particles, the number size distribution drops sharply towards the large size end. In our large data base on number size distributions measured in Vienna, concentrations of particles >1.5 µm are typically at least four orders of magnitude less than the concentrations of particles below this size.*

*To be precise: we do not measure SSA, but we nebulize (re-aerosolize) sea water samples in a Collison Atomizer, so we do not have real SSA. The droplets coming from the Collison atomizer are dried and then drawn into the CCN-VACES for activation. All droplets with sizes > 1.5µm resulting from the activation of these dry particles are then passed to the LAAPTOF*

7. Again, how to test if the separated droplets are evaporated in the vacuum before being ionized?

*Because of the construction of the LAAPTOF, such a test cannot be performed*

8. Line 275: In the negative ion spectrum signs of $O^-$ (m/z=16) and $OH^-$ (m/z=17) correspond to water. References would be helpful here. Why were these peaks not shown in every droplet?

*This interpretation is based on the paper by Neubauer et al. (1998), Neubauer et al. (1997) and Johnston (2000). In some of the individual spectra, the $O^-$ and $OH^-$ peaks are rather weak, which leads to their suppression in the central spectra obtained from the fuzzy clustering algorithm.*

9. Figure 5-7: I do not understand why these peaks are present in the negative ions spectrum.

*Does this refer to the $O^-$ and $OH^-$ peaks? The references for assigning $O^-$ and $OH^-$ to water are the studies by Johnston (2000), Neubauer et al. (1998) and Neubauer et al. (1997). In the Neubauer et al. (1997) study, the authors state that the presence of water is indicated by the presence of strong $O^-$ and $OH^-$ peaks.*

10. Line 369: "The presence of $Si^+$ (m/z = 28) in the spectrum in Fig.12 might indicate a CCN not originating in the area of Vienna". Such a statement is not satisfying and does not help in the discussion.

*The statement can be deleted. The main point here is that we saw fine Si-rich particles mainly in long-range transport air masses (Okada and Hitzenberger, 2001 ).* **done**

11. Section 5.2.2: Is there only one type of SSA? This is apparently different from previous laboratory studies.

*We analyse CCN originating from two different sea water samples – as stated above, we will state already in the introduction that we do not analyse real world SSA*

12. Section 5.2.3: I would recommend the authors focus more on what is new about this mass spectra. Does it provide more information than those for droplet residues? Only with more validation, can the author state that such a technique would provide insights into the composition of individual activated CCN.

*There is a difference between droplet residues and CCN before activation. Droplet residues have spent time in clouds and their chemical composition might have changed because of chemical reactions and / or coalescence with other cloud droplets. We look at the initial state of activated CCN. This point will be clarified in the revised MS –* ***done***